# Melatonin in Cancer Treatment: Current Knowledge and Future Opportunities

**DOI:** 10.3390/molecules26092506

**Published:** 2021-04-25

**Authors:** Wamidh H. Talib, Ahmad Riyad Alsayed, Alaa Abuawad, Safa Daoud, Asma Ismail Mahmod

**Affiliations:** 1Department of Clinical Pharmacy and Therapeutics, Applied Science Private University, Amman 11931, Jordan; a_alsayed@asu.edu.jo (A.R.A.); asmamahmod1212@gmail.com (A.I.M.); 2Department of Pharmaceutical Sciences and Pharmaceutics, Faculty of Pharmacy, Applied Science Private University, Amman 11931, Jordan; abuawad@asu.edu.jo; 3Department Pharmaceutical Chemistry and Pharmacognosy, Faculty of Pharmacy, Applied Science Private University, Amman 11931, Jordan; s_daoud@asu.edu.jo

**Keywords:** pineal gland, anticancer, cancer therapy, hormonal therapy, phytomelatonin

## Abstract

Melatonin is a pleotropic molecule with numerous biological activities. Epidemiological and experimental studies have documented that melatonin could inhibit different types of cancer in vitro and in vivo. Results showed the involvement of melatonin in different anticancer mechanisms including apoptosis induction, cell proliferation inhibition, reduction in tumor growth and metastases, reduction in the side effects associated with chemotherapy and radiotherapy, decreasing drug resistance in cancer therapy, and augmentation of the therapeutic effects of conventional anticancer therapies. Clinical trials revealed that melatonin is an effective adjuvant drug to all conventional therapies. This review summarized melatonin biosynthesis, availability from natural sources, metabolism, bioavailability, anticancer mechanisms of melatonin, its use in clinical trials, and pharmaceutical formulation. Studies discussed in this review will provide a solid foundation for researchers and physicians to design and develop new therapies to treat and prevent cancer using melatonin.

## 1. Introduction

Melatonin (N-acetyl-5-methoxytryptamine) is an indole amine produced in the human body by multiple sources. It is mainly produced by the pineal gland in response to darkness. Other organs that synthesize melatonin include skin, bone marrow, lymphocytes, retina, and gastrointestinal tract [1]

Suprachiasmatic nucleus (SCN) of the hypothalamus is the biological clock that regulates melatonin synthesis and secretion over 24 h [2]. At night, melatonin levels increase, then start to decrease in the early morning and throughout the day. Elevated levels of melatonin at night stimulate target organs to enter into suitable homeostatic metabolic rhythms which help to protect the body from the development of different diseases [3]. Therefore, exposing the body to light at night may result in disruption of melatonin production and the circadian rhythm.

Cancer is a growing health problem that needs an urgent response to control it. A prediction from the World Health Organization’s International Agency for Research on Cancer Global Cancer Observatory (GLOBOCAN) expected 27.5 million cancer cases every year by 2040. These values represent a 61.7% increase compared with current statistics [4]. Nowadays, cancer patients depend mainly on conventional anticancer therapies including surgery, radiotherapy, and chemotherapy. Additionally, many plant-derived natural products were reported to have a direct role in cancer prevention and therapy [5].

During the last decades, evidences were accumulating to support the diverse roles of melatonin in human physiology and pathology. Currently, melatonin is considered as a cell protector and not only a hormone. Studies reported essential effects of melatonin in many pathways, including oxidative stress, immune modulation, and hematopoiesis [6,7]. Additionally, a large number of studies confirmed the anticancer and oncostatic effects of melatonin mediated by different mechanisms of action [8,9]. Moreover, melatonin was used in combination anticancer therapies to augment conventional therapies and reduce side effects [8,9,10].

This review summarizes recent findings on the anticancer properties of melatonin and its mechanisms of action. Melatonin biosynthesis, bioavailability, natural sources, cancer preventive properties, pharmaceutical formulation, and its use in clinical trials were discussed in this review.

## 2. Melatonin Biosynthesis and Metabolism in Human Body

Melatonin was isolated in 1958, by the dermatologist Aaron Lerner, from bovine pineal gland. Although it is mainly secreted from the pineal gland, there are many other secondary sources including; retina, gut, skin, platelets and bone marrow, and probably other structures, but their systemic contribution is insignificant [11]. 

The starting material of melatonin biosynthesis in humans is tryptophan, an essential amino acid. Through the action of tryptophan hydroxylase (TP5H) and aromatic acid decarboxylase (AADC), enzymes tryptophan is converted to the neurotransmitter, serotonin. In the subsequent step, serotonin is converted into melatonin through the influence of arylalkylamine N-acetyltransferase (AANAT) and hydroxyindole-O-methyltransferase (HIOMT) enzymes [12] (Figure 1). 

Melatonin is not stored inside the pineal gland and it is released as it is synthetized, so the plasma hormone profile faithfully reflects the pineal activity [13]. Moreover, amphiphilic nature and the small size of melatonin facilitates its passage across cell membranes and its access to various fluids, tissues, and cellular compartments as saliva, urine, cerebrospinal fluid, preovulatory follicle, semen, amniotic fluid, and milk [11,14,15]. 

Melatonin is metabolized mainly by cytochrome P450 in the liver. It has been demonstrated that melatonin was metabolized to 6-hydroxymelatonin and N-acetylserotonin by CYP1A1 and CYP 2C19, respectively, at Phase I metabolism, and most of them were subsequently converted to sulfate conjugates by sulfotransferases in human liver and excreted in the urine [16]. A small portion of melatonin is degraded by other tissues including skin and brain by either CYPA2B or 2,3-indolamine dioxygenase to form 6-hydroxymelatonin or N^1^-acetyl-N^2^-formyl-5-methoxykynurenine (AFMK). The urinary excretion probably is not the major metabolic route of AFMK judging from its water solubility [17].

Melatonin has specific receptors to regulate many physiological functions namely; MT1 and MT2, both are members of the seven transmembrane G-protein coupled receptor family [18]. Human MT1 and MT2 receptors are 350 and 362 amino acids long, respectively, with molecular weights of 39–40 kDa and 55% amino acid homology overall [19]. Both MT1 and MT2 affect protein kinase activity through inhibition of adenylyl (cAMP) and guanylyl (cGMP) cyclase, respectively. Furthermore, activation of the phospholipase C pathway that leads to increase in inositol triphosphate (IP3) and 1, 2-diacylglycerol (DAG) levels has been proved for both MT1 and MT2 receptors [20].

## 3. Melatonin Natural Sources 

For decades, melatonin was considered an animal neurohormone. However, in 1995 studies confirmed the presence of melatonin in higher plants. Phytomelatonin is the term used to melatonin of plant origin, and today the presence of phytomelatonin is totally accepted in all plants [12,21].

Generally, the phytomelatonin content is highly variable, ranging from picograms to micrograms per gram of plant material analyzed. This variation is not appearing only among species, but also among different varieties of the same species and even between different organs of the same plant [22]. Although phytomelatonin was detected and quantified in roots, shoots, leaves, flowers, and fruits, but its highest level was found in reproductive organs, particularly in seeds [23]. It has been suggested that variations in phytomelatonin contents might result not only from the differences in extraction and detection techniques applied, but also from environmental growth conditions. As the role of phytomelatonin in plants is a protective agent against multiple stress situations, mostly higher phytomelatonin levels in plants are related with the presence of stressors, including natural, artificial, physical, or chemicals stressor [24,25].

In contrary to humans, where the melatonin precursor tryptophan can only be supplied by food, plants are able to synthesize it and theoretically it is constantly available for further transformations into phytomelatonin or other indoleamine derivatives. Besides, plants can also absorb melatonin provided exogenously from the environment and accumulating it at high concentrations and so they can be considered as rich phytomelatonin sources for humans [26]. 

Regarding the plant kingdom, first phytomelatonin was detected in the photosynthesizing alga *Lingulodinium polyedrum* in 1988, and then in vascular plants *Ipomoea nil* (L.) *Roth* (synonym, *Pharbitis nil* (L.) Choisy) in 1993 [27]. Nowadays, the occurrence of phytomelatonin in many edible plants and herbs has been widely reported. It has been found in tomatoes, cherries, olives and oils, grapes and red wine, walnuts, sunflower, mustard oil, apple, barley, cucumber, lupine, maize, rice, coffee, and medicinal plants, including pyrethrum maruna (*Tanacetum parthenium* L.) and St. John’s wort (*Hypericum perforatum* L.) [12,26]. Table 1 summarizes the content of phytomelatonin in some plants. 

Several studies demonstrated the effect of food on melatonin serum levels. Sae-Teaw et al. assessed the effect of pineapple, orange, and banana consumption on serum melatonin concentrations of healthy volunteers. Volunteers were given juice extracted from 1 kg of orange or pineapple or two whole bananas, containing 302, 150, and 8.9 ng phytomelatonin, respectively. The study demonstrated that the serum melatonin concentration was significantly increased after 120 min of fruit consumption. For pineapple (146 versus 48 pg/mL *p* = 0.002), orange (151 versus 40 pg/mL, *p* = 0.005), and banana (140 versus 32 pg/mL, *p* = 0.008), and this definitely proves that fruits are a good source of phytomelatonin. Besides, the antioxidant capacity in the serum also markedly increased, suggested by the significant increases in two indicators; ferric reducing antioxidant power assay and oxygen radical antioxidant capacity [35]. Likewise, a study conducted with young, middle-aged, and elderly participants showed that the ingestion of 200 mL of grape juice twice a day increased urinary 6-sulfatoxymelatonin, a major metabolite of melatonin commonly used as a biomarker, and total antioxidant capacity in the all three groups of individuals [36]. Besides, germination of legumes increases the plant levels of phytomelatonin, making sprouts a suitable food source of this hormone, it was reported that the melatonin concentrations in plasma increased in Sprague–Dawley rats by 16% (*p* < 0.05) after the administration of kidney bean sprout extract, which correspondingly led to the increase in urinary 6-sulfatoxymelatonin content [37]. 

## 4. Biological Activities of Melatonin 

Melatonin is widespread in nature, and it plays a vital role in different biological activities [38]. A study has been carried out in aged animals that showed melatonin’s effect on body temperature and energy balance [39]. Several studies have shed light on the melatonin immunomodulatory effect. It was reported that melatonin may regulate the activation of T/B cells in pinealectomy mice in a dose-dependent manner [40]. Besides, it shows immunomodulation and neuroprotective potential in a pharmacological Alzheimer’s disease mouse model [41]. Moreover, melatonin was known to be associated with bone homeostasis. Administration of melatonin exhibited a promising strategy to manage postmenopausal patients via restoring the osteoporosis-impaired osteogenic potential of bone marrow mesenchymal stem cells [42]. It also maintains bone balance; increases the osteogenic differentiation of bone marrow mesenchymal stem cells, and suppresses osteoclastogenesis [43]. A recent clinical trial has investigated the effect of melatonin consumption on controlling arterial pressure and anthropometric indices in type 2 diabetes mellitus patients. It reduced significantly the mean level of systolic pressure, mean arterial pressure, pulse pressure, and conicity index in the intervention group [44]. In addition, the chronobiotic properties of melatonin have been evaluated. It revealed that the administration of melatonin may regulate sleep disorders related to abnormal timing of the circadian system: jetlag, shift work, delayed sleep phase syndrome, and some elderly sleep difficulties [45]. Additionally, melatonin was able to inhibit neuroinflammation and relieve depression by autophagy modulation through FOXO3a signaling [46]. Recently, melatonin has been investigated as a candidate drug for the management of corona virus infection. It docks with novel coronavirus proteins and exhibits a variety of interactions with an interesting docking score that leads to prevent the virus proteins, which lead to demolish the virus as well [47]. 

## 5. Melatonin as Antioxidants

The oxidative stress refers to the physiological disturbance between the production of reactive oxygen species (ROS) and the ability of the body to remove them [48]. A variety of ROS are generated during a number of processes, such as inflammation, infection, mechanical and chemical stresses, and exposure to UV rays and ionizing radiation [49]. Although the basal levels of ROS act as signaling molecules to activate cell proliferation, survival, apoptosis, differentiation, and immune response pathways, the high levels of ROS can damage DNA, protein, and lipids which lead to mutations and promote carcinogenesis [50]. The human body can counteract the oxidative stress by antioxidants, which are either naturally produced in situ (endogenous) or externally supplied through foods and supplements (exogenous), and, therefore, enhance the immune defense and lower the risk of disease and cancer [51]. 

Although the main physiological functions of melatonin are related to hormonal properties, it has been linked to a wide range of functions. One of these essential functions is its ability to act as antioxidant [52]. 

An electron rich aromatic system and the amphiphilicity of the compound arising from O-methyl and N-acetyl residues are supposed to be the molecular bases for its antioxidant properties [53]. Melatonin’s antioxidant properties involved many mechanisms. It can directly scavenge ROS and reactive nitrogen species (RNS) [54]. For instance, it was reported that in in vitro the ability of melatonin to scavenge the hydroxyl radical (·OH) was much higher compared with that of vitamin E, which is the reference in the field [11]. Moreover, melatonin can regulate the activities of several antioxidant enzymes like superoxide dismutase, glutathione reductase, glutathione peroxidase, and catalase [55]. Studies also proved the ability of melatonin to stimulates the synthesis of other antioxidants. For instance, melatonin was found to induce the expression of gamma-glutamylcysteine synthetase (γ-GCS), the rate-limiting enzyme of glutathione (GSH) synthesis [34]. Melatonin can also increase the efficiency of the mitochondrial electron transport chain, thereby easing electron leakage and, thus, reducing the generation of free radicals [56]. 

Under the condition of severe oxidative stress, melatonin is found to be metabolized via enzymatic degradation or free radical interactive processes to produce many metabolites including hydroxylated melatonin metabolites (6-hydroxymelatonin, 2-hydroxymelatonin, and 4-hydroxymelatonin), N1-acetyl-N2-formyl-5-methoxykynuramine (AFMK), N-acetyl-5-methoxyknuramine (AMK), and cyclic 3- hydroxy melatonin. Interestingly, all these metabolites can also act as antioxidants. Consequently, melatonin can generate a radical scavenger cascade with never ending action even with its degradation metabolites. This cascade predictably allows melatonin to neutralize up to 10 radical products, which contrasts with classic free radical scavengers, which detoxify a single oxidizing molecule [57]. Therefore, unlike other antioxidants, melatonin is very effective even in small doses [53,58,59]. Surprisingly, some of the metabolites are even more potent than its precursor. For example, AMK showing stronger capability of scavenging ROS and preventing protein oxidation than its precursor [60]. More and more, melatonin can enhance the activity of other antioxidants; Xu et al. reported that melatonin treatment enhanced the polyphenol content and antioxidant capacity of red wine [61]. Similarly, the protective effects of vitamin E, glutathione, or vitamin C against iron-induced lipid peroxidation were dramatically enhanced with melatonin combination [62].

Accordingly, melatonin could be an excellent candidate for the prevention and treatment of several cancers, such as breast cancer, prostate cancer, gastric cancer, and colorectal cancer [2]. It was demonstrated that 1 mM melatonin concentration is the pharmacological concentration that is able to produce anticancer effects [63,64,65]. Goncalves et al. revealed that the melatonin at 1 mM concentration can induce an anti-metastatic effect on MCF-7 breast cancer cell line through inhibiting the viability and invasiveness of the cancer cells [66]. Furthermore, melatonin at 1 mM concentration can inhibit the angiogenesis in MCF-7 breast cancer cells; Alvarez-Garcı’a et al. suggest that melatonin may play a role in the paracrine interactions between malignant epithelial cells and proximal endothelial cells through a downregulatory action on VEGF expression in human breast cancer cells, which decrease the levels of VEGF around endothelial cells [67,68]. Likewise, melatonin mediates the anti-angiogenic property in hypoxic PC-3 Prostate Cancer cells by upregulation of microRNA3195 and microRNA374b [53]. Similarly, Lv et al. reported that 1 mM concentration of melatonin exhibited high inhibitory effect on cellular proliferation of pancreatic carcinoma cells (PANC-1), along with a significantly decrease in vascular endothelial growth factor (VEGF) [69]. On the other hand, many studies confirmed that melatonin at physiological concentration (1 nM) exerts antiproliferative actions and induces apoptosis in breast, prostate, and ovarian cancer cells [70]. Figure 2 summarizes the mechanisms of action of melatonin in reducing oxidative stress.

## 6. Melatonin and Cancer Hallmarks

### 6.1. Role of Melatonin in Maintaining the Genomic Integrity of Cells

Genomic instability is one of the factors that led to tumorigenesis in cells. In normal mammalian cells, different pathways were followed to preserve genomic stability [71]. It is summarized in four major mechanisms: high-fidelity DNA replication in the S-phase, accurate chromosome distribution during mitosis, flawless repair of sporadic DNA damage, and progression of the cell cycle [72].

The antioxidant activity of melatonin plays an important role in protecting DNA from oxidative damage. It is either directly through scavenging of free radicals, or indirectly by inhibiting metal-induced DNA damage, stimulating antioxidant enzymes, enhancing the DNA repair system, and inhibiting pro-oxidative enzymes [73]. Based on a recent study, tumor-bearing mice were treated with (10 mg/kg) of melatonin and (5 mg/kg) of cisplatin. The results have shown that melatonin was able to reduce DNA damage and improve the anticancer effect of cisplatin [74]. Moreover, melatonin (1200 μg/mL) was able to reduce the production of micronucleus induced by γ-radiation in two cell lines: HeLa and MRC5 cells. Its scavenging effect and stimulation of DNA repair pathways were significantly reported [75]. Besides, melatonin attenuated the cytotoxic effect of arecoline in oral squamous cell carcinoma (OSCC) by activating antioxidant enzymes and protecting the DNA integrity. It also enhanced arecoline-induced ROS formation, G2/M phase arrest, and cell apoptosis [76]. Another study has shown the effect of melatonin pretreatment in breast cancer cells (MCF-7) one week before radiation exposure. It reduced cell proliferation, promoted cell cycle arrest, and resulted in a remarkable decrease in RAD51 and DNA-PKcs mRNA expression compared to the free melatonin cells [77].

Many previous studies have reported the effects of melatonin on mitochondrial function by decreasing ATP production, activating the reactive oxygen species (ROS) to promote apoptosis in cancer cells, and inhibiting telomerase activity [1,78,79,80].

### 6.2. Melatonin Effect on Proliferative Signaling

Cancer cells are recognized by their ability to over-proliferate via modulation of protein expression and signaling pathways. The most critical and controlling pathways are hypoxia-inducible factor-1 (HIF-1), NF-κB s, PI3K/Akt, insulin-like growth factor receptor (IGF-1R), cyclin-dependent kinases (CDK), and estrogen receptor signaling [81].

Melatonin combined with 5-fluorouracil showed antiproliferation, antimigration, and proapoptotic effects in colon cancer cells by downregulation of PI3K/AKT and NF-κB/iNOS signaling pathways [82]. Another study has reported that melatonin (3.4 Mm) inhibited proliferation of CSCs (cancer stem cells isolated from ovarian cancer cells), and reduced protein expression of Ki67 and matrix metalloproteinase 9 (MMP9). It also prevents cell migration via alteration of PI3K and MAPK signaling pathways in both receptor-dependent and independent manner [83]. Melatonin was able to suppress the cell growth of breast cancer cells (MDA-MB-231) in both in vitro and in vivo. The results showed significant downregulation of the HIF-1α gene and protein expression coupled with the production of GLUT1, GLUT3, CA-IX, and CA-XII [84].

In hypoxic PC-3 cells (prostate cancer cells), melatonin could reduce HIF-1α accumulation by inactivation of SPHK1 (new modulator of HIF-1α) [85]. Many studies have shown the inhibitory effect of melatonin against nuclear factor-kappaB (NF-κB) in various cancer cell lines such as lung cancer and liver cancer [86,87,88]. Moreover, upregulation of p21, p27, and PTEN protein is another way of melatonin to promote cell programmed death in uterine leiomyoma growth. It also reduced tumor growth in both xenograft and orthotopic uterine tumor mice models [89]

Furthermore, melatonin (4 mmol/L) has suppressed cell proliferation in human osteosarcoma cells (MG-63 cells) via reducing expression of cyclin D1, CDK4, cyclin B1, and CDK1 [90].

### 6.3. Melatonin Effect on Promoting Cell Apoptosis

The loss of apoptotic control is one of the cancer hallmarks. It allows unlimited growth, promotes angiogenesis, and apoptosis evasion [91,92]. B-cell lymphoma-2 (Bcl-2) family proteins composed of members that control apoptosis. Moreover, activation of the pro-apoptotic pathway proteins is one of the main targets to conquer cancer cells [93]. According to the literature, melatonin has increased the expression of pro-apoptotic mediators such as BAX/BAK, Apaf-1, caspases, and p53 [94]. Further, treating gastric cancer cells with melatonin (2 mM) resulted in stimulating apoptosis via increase expression of p38 and p-JNK protein, and downregulation of p65. JNK and p38 are associated with cell growth and apoptosis, while NF-κB-p65 signaling pathway mediated suppression of apoptosis [95]. Melatonin augmented the cisplatin activity in human cervical cancer cells by inducing caspase-9-dependant mitochondrial apoptosis, and increasing pro-apoptotic proteins production [96]. Another study had shown the ability of melatonin to inhibit Bcl-2 and upregulate BAD/BAX genes in MCF-7 human breast cancer cells [97]. Melatonin was also able to induce apoptosis in colorectal cancer cells via an increase in superoxide production and suppress cellular prion protein expression [98]. In human gastric cancer cells, melatonin was able to enhance BAX expression, reduce Bcl-xL production, activate caspase-3 and 9, and inhibit the AKT/MDM2 intracellular pathway [99]. According to Yu et al., the anti-apoptotic activity of melatonin against neural cancer cells was due to its metabolite N-acetylserotonin (NAS) resulted from mitochondrial cytochrome P450 (CYP) 1 B1 overexpression [100]. Besides, melatonin showed an anti-apoptotic effect on several types of cancer cells with different mechanisms [100,101,102].

### 6.4. Melatonin Effect on Angiogenesis Process

Angiogenesis is the formation of new blood vessels. This new vascular network is very important for cancer cells, as it helps to establish cell proliferation and metastatic spread [103]. Many factors control angiogenesis, such as vascular endothelial factor (VEGF), platelet-derived growth factor (PDGF), epidermal growth factor (EGF), and hepatocyte growth factor (HGF) [104]. A study has shown the activity of melatonin in inhibiting VEGF expression in SH-SY5Y human neuroblastoma cells [105]. Additionally, the anti-angiogenic effect of melatonin was observed in breast cancer cells (MDA-MB-231). It was able to reduce the gene level of miR-148a-3p, IG-IR, and VEGF, both in vitro and in vivo [106]. On the same type of cancer, but different cell line (MDA-MB-468), melatonin has shown attenuation effect on miR-152-3p expression and suppression of IGF-1R, HIF-1α, and VEGF production [107]. Moreover, a reduction in VEGF expression was reported in ovarian cancer cells (SKOV3) treated with melatonin [108]. A recent study has revealed that melatonin was able to modulate tumor angiogenesis in osteosarcoma by upregulation of miR-424-5p expression and inhibiting of VEGF [109]. Besides, melatonin prevented angiogenesis in HepG2 liver cancer cells via inhibition of HIF-1α and STAT3 signaling pathway [63].

### 6.5. Role of Melatonin in Tumor-Associated Immune Evasion

There are several mechanisms by which tumors evade an immune response including activation of regulatory cells, deformed antigen presentation, immune suppression, and immune deviation [110]. Melatonin has an impact on immune cells by enhancing their viability, improving cell metabolism in the tumor microenvironment, and modulating cytokines release [111]. It was reported that melatonin can stimulate T cell and natural killer (NK) production in in vivo study [112]. Liu et al. demonstrated the reduction effect of melatonin on regulatory T cells (Tregs) and Forkhead box p3 (Foxp3) in gastric cancer cells [113]. According to Wongsena et al., melatonin has an immunomodulatory effect in hamsters treated with a chemical carcinogen. The results showed that a dose of 50 mg/kg melatonin was able to reduce eosinophils, Th17 cells, and Foxp3 expression, as well as increase CD4+ and TNF-α accumulation [114]. Other studies exhibited stimulation of Th1 in tumor-bearing mice treated with melatonin [10,115].

### 6.6. Role of Melatonin in Tumor-Promoting Inflammation

There is a strong relationship between chronic inflammation and tumor development [116]. The outcomes of the inflammatory process might promote carcinogenesis via the formation of reactive oxygen species and reactive nitrogen species which play a role in DNA damage and cancer development [117]. Melatonin has shown an anti-inflammatory effect in human intestinal cells stimulated by interleukin-1β. As a result, the inflammatory mediators (IL-6, IL-8, COX-2, and NO) have been downregulated significantly along with suppression of NF-κB expression and protecting DNA from damage [118]. Another study reported that melatonin can modulate ER stress-associated TXNIP/NLRP3 inflammasome activity in LPS-induced endometritis in mice [119]. The antitumor effect of melatonin was augmented by inducing Bim expression and reducing COX-2 expression in tunicamycin-induced apoptosis breast cancer cells [120].

### 6.7. Role of Melatonin in Tumor Dysregulated Metabolism

Cancer cells tend to convert most glucose to lactate, even in the presence of oxygen. This is known as (the Warburg effect). This pathway enhanced tumor cells to synthesize macromolecules required for rapid cellular proliferation, reduced cell apoptosis, and eventually provided a suitable environment for tumor to metastasis [121,122]. It was suggested that melatonin stimulate the synthesis of acetyl-CoA from pyruvate by inhibiting the mitochondrial enzyme pyruvate dehydrogenase kinase (PDK) [123]. A study has shown that melatonin has altered Ewing sarcoma metabolic profile by inhibiting the Warburg effect [124]. In prostate cancer cells, melatonin was able to reduce glucose metabolism via downregulation of glycolysis, tricarboxylic acid cycle, and pentose phosphate pathway [125].

### 6.8. Melatonin Effect on Tissue and Metastasis

Cancer metastasis means that cancer cells spread away from the original tumor to surrounding tissues and distant organs. Multi-steps are involved in metastasis, including infiltration of tumor cells into the adjacent tissue, intravasation, traveling in the circulatory system, extravasation, and overproliferation in the competent organs [126]. Melatonin exhibited various mechanisms to restrain metastasis, such as modulation of cell–cell and cell–matrix interaction, extracellular matrix remodeling by matrix metalloproteinases, readjustment of the cytoskeleton, epithelial-mesenchymal transition, and angiogenesis [127]. A recent study reported that the overexpression of the ADAMTS family in renal cell carcinoma was suppressed by melatonin via amplifying of miR-let-7f/miR-181d and reducing protein stability. It is of note that ADAMTS, a disintegrin, and metalloprotease with thrombospondin motifs family, thought to have an impact on cell metastasis and developing of cancer stages [128]. Based on the Bu et al. results, melatonin inhibits chronic restraint stress-mediated metastasis of epithelial ovarian cancer by reducing the NE/AKT/β-catenin/SLUG axis [129]. Another mechanism of melatonin was inhibiting of matrix metallopeptidase 13 (MMP-13) in prostate cancer cells in both in vitro and in vivo [130]. Moreover, it was found that melatonin can hinder cancer-associated osteoclast differentiation via modulation of tumor-secreted RANKL expression in lung and prostate bone metastasis models [131]. Gu et al. revealed the anti-metastasis effect of melatonin against esophageal cancer. There was a reduction in MMP-9 expression along with a high level of E-cadherin and inhibition of the NF-κB signaling pathway [132]. Besides, the migration of human lung adenocarcinoma cell line A549 was suppressed after using melatonin. This inhibition is mediated by an increase in occludin expression [133]. Additionally, applying melatonin in breast cancer cells caused a decrease in the expression of vimentin and inhibition of cell migration [66]. It was also reported the inactivation of MMP-2 and MMP-9 signaling pathway in breast cancer cells treated with melatonin [134]. Figure 3 summarize the role of melatonin in cancer hallmarks.

## 7. Melatonin Bioavailability and Use in Cancer Treatment

### 7.1. The Use of Melatonin in Cancer Treatment

Plethora of clinical research have reported the oncostatic role of melatonin against various types of cancer such as Gastric cancer [135,136,137], breast cancer [67,138,139], oral cancer [140,141], prostate cancer [142,143,144], and other more types.

#### 7.1.1. Gastric Cancer

Gastric cancer (stomach cancer) is one of the most common cancers worldwide. According to GLOBOCAN 2018 data, gastric cancer is the 3rd most deadly cancer [145]. Melatonin has been reported with distinguished anticancer activity against gastric cancer. The anti-gastric cancer mechanisms of melatonin are still not fully understood, however, various studies suggested several mechanisms for anticancer activity of melatonin including stimulation of immunity, cell proliferation inhibiting, and apoptosis induction [146,147]. Zhang et al. have investigated the impact of melatonin on the functions of gastric adenocarcinoma cell line, SGC7901, including apoptosis, cell proliferation, cell migration, and colony formation. They demonstrated that melatonin could inhibit colony formation, cell proliferation, cell migration, and enhanced apoptosis [135]. In another study on SCG7901 human gastric cells, Wang et al. have illustrated the association of melatonin with RZR/RORγ pathway under hypoxia. Their results showed suppression in the activity of RZR/RORγ, in addition to suppression in SUMO-specific protease 1 (SENP1) signaling pathway, which is crucial for stabilizing the hypoxia inducible factor-1α (HIF 1α) during hypoxia in response to melatonin. Moreover, melatonin was able to reduce the vascular endothelial growth factor (VEGF) expression and suppress metastasis [136]. In agreement with this, Wang et al. followed up with another study to evaluate the anticancer activity of melatonin on the growth and angiogenesis of SGC7901cells, revealing the inhibitory effect of melatonin on the growth of SGC7901cells. The low concentration of melatonin (0.01, 0.1, and 1 mM) had no clear impact on VEGF secretions, however, higher concentration (3 mM) had clearly suppressed VEGF secretions. Besides, the expression of melatonin nuclear receptor RZR/RORγ, HIF-1α, SUMO-specific protease 1, and VEGF had been reduced within SGC7901 during tumorigenesis in response to treatment with melatonin [148]. In addition, Song et al. have investigated the effect of melatonin on SGC7901 cells in term of protein production using protein chip technology. Melatonin was found to induce cell cycle arrest. Furthermore, melatonin induced changes in proteins that are related to cell proliferation and apoptosis represented in downregulation of phospho-CDC25A, CDC25A, p21, phosphor-p21, and Bcl-xl, upregulation of Bax, an activation of caspase-3 and an increase in the level of cleaved caspase-9, which ensured the implication of mitochondria in melatonin-induced [99].

#### 7.1.2. Glioblastoma

Glioblastoma is the most common and aggressive primary brain tumor in adults. The incidence rate of glioblastoma is 5–8 per 100,000, representing around 54% of diagnosed gliomas cases. Glioblastoma has short life expectancy, less than one year since diagnosis in average, which is owed to the tumor recurrences high rate [149,150]. Glioblastoma was reported with higher frequency and 1.6 higher incidences in males as compared to females [150,151]. Glioma stem-like cells are subpopulation in glioblastoma, they play a crucial role in the tumor growth maintenance and recurrence [152,153,154], and promote self-renewing capacity and tumor propagation [155,156,157]. Melatonin showed an anticancer effect against glioblastoma, and it was also reported to overcome the multi-drug resistance in glioblastoma treatment [158,159,160]. Sung et al. recently have investigated the impact of combination of melatonin with vorinostat on the expression of transcription factor EB and apoptosis in glioblastoma cells and glioma cancer stem cells. The expression of transcription factor EB, which needs oligomerization to regulate transcription, was reported to be increased in glioblastoma. The combination of vorinostat and melatonin induce a downregulation of the transcription factor EB and oligomerization, which increased apoptosis related gens, hence, cells apoptosis was activated [161]. In another study, Chen et al. have studied the roles of melatonin and the associated mechanisms against glioblastoma stem-like cells. Their results demonstrated that melatonin altered the glioblastoma stem-like cells biology and inhibited glioblastoma stem-like cells proliferation. Moreover, melatonin showed to alter the transcription factors profile inhibiting the initiation and propagation of tumor. In addition to the impairment of EZH2–STAT3 interaction and EZH2 S21 phosphorylation, melatonin has multiple roles in attenuating several key signals related to survival and self-renewal in glioblastoma stem-like cells [158]. Lai et al. have studied the microenvironment of glioma investigating the correlation of melatonin treatment and molecular markers in glioblastoma multiform including SIRT1, CCL2, ICAM-1, and VCAM-1. Their results showed melatonin administration increased the expression of SIRT1, which inhibit the growth and proliferation of glioma cells [162]. In another recent study, Fernandez-Gil et al. have explored whether treatment with melatonin can restore the oxidative phosphorylation after metabolic switch to glycolysis in glioblastoma cells. The results showed that melatonin significantly decreased the viability and inhibited the proliferation of glioblastoma cells. Besides, it modulates a metabolic shift from glycolysis to oxidative phosphorylation, which lead to a reduction in the malignant properties of glioblastoma cells [163]. Additionally, it was reported that the melatonin antitumor effect can be through suppression of the EZH2-NOTCH1 signaling axis in glioblastoma stem-like cells [164]. Moreover, several studies have shown the melatonin impact on glioblastoma cells via enhancing apoptosis and inhibiting cell migration and invasion [165,166,167].

#### 7.1.3. Prostate Cancer

Prostate cancer (PC) is the most common cancer in males. It is the fifth leading cause of death in men cancer cases worldwide [168,169]. Prostates represent a target for melatonin which has been proven with its inhibitory effect on the cell growth of prostate cancer [170,171,172]. Wang et al. have investigated the effect of melatonin on prostate cancer cells. Their results showed that melatonin downregulated the expression of matrix metallopeptidase 13 (MMP-13) and inhibited the invasive and migratory capacities in prostate cancer cells via the phospholipase C, p38, and c-Jun signaling cascades and MT_1_ receptor. MMP-13 have been reported to be highly expressed in prostate cancer patients as compared to healthy individuals. Moreover, melatonine suppressed the growth rate and metastasis in prostate cancer cells in both in vivo and in vitro models [130]. In a retrospective study, Zharinov et al. have evaluated the use of melatonin in prostate cancer patients with different risk groups showing that there is no significant difference between the melatonin-treated and not treated in the favorable and intermediate prognoses groups. However, an increase in the survival rate in poor prognosis group has been demonstrated in melatonin-treated patients as compared to untreated patients [173]. Liu et al. investigated the melatonin activity in 22Rv1 and LNCaP prostate cancer cells. They showed that these cells overexpress androgen receptor splice variant-7 (AR-V7) and activate nuclear factor-kappa B (NF-κB) that results in upregulation of the expression of IL-6. Melatonin showed inhibitory effect on expression of AR-V7 and its induced activation of NF-κB and IL-6 gene transcription [174]. Besides, Guilherme et al. have evaluated the activity of melatonin alone or combined with docosahexaenoic acid on PNT1A prostate cancer cells in regard to proliferation relevant pathways, ROS production, and mitochondria bioenergetics. Melatonin upon coincubation with docosahexaenoic acid improved the oxidative phosphorylation and restored mitochondrial bioenergetic reserve capacity. These melatonin induced alterations were related to AKT/mTOR dephosphorylation, and modulation of ERK1/2 expression [175]. An in vivo study has demonstrated the antitumor effect of melaonin on prostate cancer [144]. Moreover, melatonin inhibited angiogenesis in prostate cancer cells via amplifying the miRNA3195 and miRNA374b expression [68]. It also inhibited cell growth in LNCap and PC-3 cell line [142].

#### 7.1.4. Lung Cancer

In cancer related-deaths globally, lung cancer is one of the most common type that is well known with its strong metastasis [176,177]. It is the second most common cancer in males and females according to the American Cancer Society (ACS) [178]. Melatonin has shown its effectiveness against lung cancer [179,180,181]. Recently, Ma et al. have studied the effect of melatonin on non-small cell lung cancer. Melatonin administration remarkably enhanced apoptosis, in addition to inhibition of proliferation, invasion, and metastasis in NSCLC. In addition, melatonin reduced the level of HDAC9 in NSCLC [179]. In another study, Yun et al. have investigated the effect of administration of melatonin in combination with gefitinib in H1975 NSCLC and HCC827 lung tumor cell line. The results showed that co-administration of melatonin with gefitinib reduced the viability of H1975 cells with harbored T790M somatic mutation, as compared to HCC827 cells with an active epidermal growth factor receptor (EGFR) mutation. This decreased viability and cell death lead to reduced phosphorylation of EGFR and Akt, in turn, decreasing the expression of several survival proteins; such as Bcl-xL, Bcl-2, and surviving, and activating caspase 3 in H1975 cells. Additionally, it was found that co-administration induced apoptosis and downregulated EGFR phosphorylation in H1975 as compared to administration of melatonin or gefitinib alone, suggesting that melatonin acts by increase the sensitivity of H1975 cells to gefitinib [180]. Furthermore, Plaimee et al. have evaluated the anticancer effect of melatonin in combination with cisplastin in SK-LU-1, human lung adenocarcinoma cisplatin-sensitive cell line. The results showed that co-administration of melatonin decreased the IC50 of cisplatin and enhanced apoptosis of SK-LU-1 cells via increasing the membrane polarization of mitochondria, activating caspases-3/7, and promoting cell cycle arrest, as compared to using cisplatin alone [181]. Besides, Zhou et al. have explored the anticancer effect of melatonin and its mechanism on A549 cells, human lung adenocarcinoma cell line. Treatment with melatonin decreased the viability and inhibited migration of A549 cells. Moreover, downregulation of the expression of MLCK and OPN have been observed, in addition to a reduction in phosphorylation of MLC of A549 cells. However, an elevation in the occludin expression involving JNK/MAPK pathway have been demonstrated suggesting that these effects mediate inhibition of the migration of A549 [133].

#### 7.1.5. Ovarian Cancer

Ovarian cancer is the main cause of death worldwide among the gynecological malignancies [182]. Melatonin has been reported with its efficiency against ovarian cancer [183,184]. Chuffa et al. have studied the anti-inflammatory activity of melatonin in modulation of toll-like receptors (TLR) which expressed on the surface of ovarian cancer. The results showed that there is no decrease in the level of TLR2 in response to melatonin. However, the ovarian cancer-associated increase in several proteins was suppressed by melatonin. Moreover, melatonin decrease the expression of IRF-3, IkBα, TRIF, p65, and NF-kB, which are involved in TLR4 mediated signaling pathway, suggesting the role of melatonin in attenuating the TLR4-mediated TRIF- and MyD88-dependent signaling pathways in ovarian cancer in ethanol-consuming rats [183]. Akbarzadeh et al. also explored the cytotoxic activity of melatonin alone or in combination with photodynamic irradiation on HUVEC umbilical cells and SKOV3 ovarian cancer cell line. A remarkable increase in the levels of reactive oxygen species generation, apoptosis–necrosis rate, and heat shock protein 70 expression was reported in both cell lines in response to the combination of melatonin and photodynamic therapy. This can highlight the melatonin as an enhancing agent for the apoptosis and efficacy of laser therapy in ovarian cancer cells [185]. In another recent study, ZemŁA et al. have explored the effectiveness of using melatonin with the anticancer drug, cisplatin on SK-OV-3, IOSE 364, and OVCAR-3 ovarian cancer cell lines. This study demonstrated that melatonin at certain concentration showed synergistic effect with cisplatin. Moreover, this synergism found to be independent of membrane melatonin receptor MTI [186]. Ataei et al. have explored the activity of melatonin as inhibitor for the Cadmium-induced proliferation in SK-OV-3 and OVCAR-3 cell lines. While cadmium showed proliferation enhancement, melatonin showed inhibition of this cadmium-induced proliferation. Furthermore, melatonin inhibited the cadmium-induced effect on estrogen receptor α expression in SK-OV-3 and OVCAR-3 cells [184]. A study has demonstrated the effect of melatonin in ovarian cancer cells (OVCAR-429 and PA-1). It repressed cell growth and downregulated CDK2 and 4 [187]. Interestingly, using long-term treatment of melatonin in an in vivo model of ovarian carcinoma (OC), exhibited high potency of melatonin in regulating different signaling pathways associated with OC [188].

#### 7.1.6. Colorectal Cancer

Colorectal cancer is a challenging cancer, with a high expected incidence in elderly people. Its signs and symptoms depend on the anatomical location, tumor progression, and cancer stage [189,190,191] However, 60% of cases can be monitored with therapies [192]. Melatonin has been used as an anticancer therapy in colorectal cancer [193,194]. Wang et al. have investigated the effect of combining melatonin with ionizing radiation on HCT 116 human colorectal cancer cell line in vitro and in vivo Melatonin inhibited proliferation, cell migration, and colony formation in HCT 116 following ionizing radiation. This increase in radiosensitivity of the cells was in association with cell cycle arrest in the phase G2/M, activation of caspas-related apoptosis, and decrease in the expression of proteins involved in break repair. In vivo, cell growth of the xenografted tumor was significantly inhibited after treatment with melatonin and ionizing radiation as compared to each agent alone, hence, higher tumor suppression rate suggesting melatonin sensitizing the colorectal cancer cells in cancer radiotherapy [194]. In an attempt to explore apoptosis activity of melatonin, Wei et al. have investigated the mechanism of melatonin-induced apoptosis in LoVo colorectal cancer cell line. It was found that melatonin inhibited proliferation and promoted apoptosis in LoVo cells. It was observed that the melatonin induced apoptosis via nuclear import and dephosphorylation of histone deacetylase 4 (HDAC4), as well as reduced the expression of Bcl-2 [193]. In another study, Yun et al. have explored the apoptic and the pro-oxidant effect of melatonin in wild type human colorectal cancer cell line (SNU-C5/WT). It was found that melatonin increased the production of superoxide via decreasing the levels of PTEN-induced kinase 1 (PINK1) and cellular prion protein (PrP^C^). This induces endoplasmic reticulum stress and apoptosis. The results of this study have shed the light on a promising targeting strategy in colorectal cancer [98]. In the same line, Lee et al. have investigated the PrP^C^ level in oxaliplatin-resistant colorectal cancer (SNU-C5/Oxal-R). Significantly increased levels of PrP^C^ was found in SNU-C5/Oxal-R as compared with SNU-C5/WT colorectal cancer. Interestingly, co-administration of melatonin with oxaliplatin downregulated the PrP^C^ expression and increased the superoxide production. Moreover, apoptosis and endoplasmic reticulum stress were remarkably increased in SNU-C5/Oxal-R following co-administration of melatonin with oxaliplatin suggesting the role of as a key protein in resistance to oxaliplatin in SNU-C5/Oxal-R [195]. Antitumor activity of melatonin was also reported in human colorectal cancer cells (HCT116). Melatonin amplified apoptosis action, autophagy, and senescence in cancer cells [196]. Besides, it was able to prevent cell migration in RKO colon cancer cells via suppression of ROCK expression [197].

#### 7.1.7. Oral Cancer

Oral cancer is a highly aggressive cancer with a high mortality rate worldwide [198]. Chemotherapy showed beneficial activity for survival in local oral cancer [199]. Liu et al. have investigated the effect of melatonin on SCC9, SCC25, Cal27, Tca8113, FaDu, and hNOKs oral cancer cells. It was found that the apoptosis resistance and proliferation were impaired upon treatment with melatonin. This effect was due to inactivation of ROS-dependent Akt signaling, downregulation of Bcl-2, PCNA, and cyclin D1. Melatonin also decreased invasion and migration of oral cancer cells [200]. Yeh et al. have explored the antimetastatic activity of melatonin in OECM-1 and HSC-3 oral cancer cell lines. Their results demonstrated that melatonin hampered the migration of OECM-1 and HSC-3 cells; in addition, it decreased the activity of MMP-9 enzyme, as well as its expression of mRNA and protein. Moreover melatonin showed a suppression effect on the phosphorylation of the ERK1/2 signaling pathway that decreased the gene transcription of MMP-9 [201]. Additionally, Yang et al. have evaluated the action of melatonin on oral cancer patient-derived tumor xenograft as a model and in oral squamous cell carcinoma. They examined the effect of overexpressing of histone lysine-specific demethylase (LSD1). Melatonin significantly suppressed the cell proliferation of oral squamous cell carcinoma in a time- and dose-dependent manner. The results suggested that proliferation suppression was associated with melatonin-induced inhibition of histone lysine-specific demethylase in oral cancer in vitro and in vivo [202]. In a recent study, Hunsaker et al. have evaluated the effect of melatonin on the microRNA content in the extracellular vesicles in different oral cancer cell lines including CAL27, SCC25, and SCC9. The results showed differential effect of melatonin on specific microRNAs in the three oral cancer cell lines highlighting the importance of evaluation of microRNA when studying the anti-oral cancer activity of melatonin [203]. Another study has shown the effect of melatonin on suppressing molecular proteins associated with angiogenesis and metastasis in oral carcinoma cells [141]. Besides, antiapoptotic activity of melatonin was reported in VCR-resistant oral cancer cells [204].

#### 7.1.8. Liver Cancer

Liver cancer is the fourth leading cause of cancer death globally in 2018 [169]. Several studies have reported the efficiency of melatonin against hepatocarcinoma cells [63,146]. Ordoñez et al. have evaluated the role of melatonin in ceramides metabolism and autophagy in HepG2 cells, human liver cancer cell line. Melatonin promoted autophagy in HepG2 cells via JNK phosphorylation which is characterized by an increase in p62 degradation, Beclin-1 expression, and colocalization of LAMP-2 and LC3II that lead to decreased cell viability. Furthermore, melatonin increased the ceramides levels through acid sphingomyelinase (ASMase) stimulation and de novo synthesis indicating. Given the crucial role of ceramides in regulating the autophagy, it is indicated the effect of melatonin on autophagy and apoptosis through affecting the ceramides metabolism [205]. Carbajo-Pescador et al. have investigated the anti-angiogenic activity of melatonin in HepG2. It was found that melatonin decreased the levels of VEGF, the expression of HIF-1α protein under hypoxic conditions. Furthermore, melatonin inhibited the hypoxia-induced increase in phospho-STAT3, CBP/p300, and HIF-1α and inhibited their physical interaction, suggesting that melatonin exhibited its anti-angiogenic effect by interfering with VEGF transcriptional activation through HIF-1α and STAT3 [63]. Cheng et al. have evaluated the effect of melatonin on the exosome derived from hepatocarcinoma cells and the expression of inflammatory factors. Melatonin reduced the expression of programmed death ligand 1 on macrophages. Furthermore, melatonin inhibited the high expression of the inflammatory cytokines; TNFα, IL-10, IL-6, and IL-1β in macrophages. It was found that exosomes derived from melatonin treated hepatocarcinoma cells can change the immunosuppression state via STAT3 axis in macrophages, suggesting the role of melatonin in manipulating the immunosuppressive state in hepatocarcinoma cells [206]. Besides, melatonin has reduced the expression of HIF-1α, VEGF, and suppressed cell proliferation in hepatocarcinoma cells [207]. In addition, Human hepatoma cell apoptosis has been induced by melatonin via downregulation of COX-2 [208].

#### 7.1.9. Renal Cancer

Several studies have focused on the role of melatonin as an anticancer in renal cancer [209,210]. Abraham et al. have explored whether melatonin can prevent Methotrexate-induced renal damage in rats. The results revealed that the rats which were treated with melatonin prior to methotrexate treatment showed a reduction in methotrexate-induced renal damage biochemically and histologically. Moreover, pretreatment of melatonin showed a reduction in Methotrexte-induced oxidative stress and perturbation in the antioxidant enzymes, indicating the beneficial role of melatonin in decreasing the Methotrexate-induced side effects in renal cancer cells and tissues [211]. Park et al. have investigated the mechanism underpinning melatonin effect on renal cancer Caki cells. It was shown that melatonin promoted apoptosis; it elevated the proapoptic protein Bcl-2-interacting mediator of cell death (Bim). Melatonin increased the mRNA expression of Bim through increasing the expression and transcriptional activity of E2F1 and Sp1, suggesting that melatonin promotes apoptosis in renal cancer Caki cells via increasing Bim expression [212]. Recently, Lin et al. have studied the impact of melatonin on the migration and invasion of Caki-1 and Achn renal cancer cell lines. Melatonin inhibited migration and invasion of these cells. Furthermore, melatonin decreased MMP-9 by decreasing p52- and p65-DNA-binding activities. In addition, ERK1/2 and JNK1/2 signaling pathways were implicated in the melatonin regulatory effect on cell motility and MMP-9 transactivation, indicating the impact of melatonin on motility and metastasis of renal cancer cells [213]. Table 2 summarizes the anticancer effect of melatonin on different cancer types with the mechanisms of action.

### 7.2. Bioavailability of Melatonin

Exogenous melatonin is being exceedingly used for treatment of various pathological conditions and diseases [221,222]. However, its pharmacokinetic properties are still not fully understood [223], so neither the optimal dose to elicit a therapeutic effect [224]. Few experimental studies have been applied to compare the bioavailability of oral and intravenous melatonin in humans, also these studies are different in the number of the subjects, dose regimen, and pharmacokinetic analytical methods, in addition to the scarcity of studies that include intravenous formulations of melatonin [224,225]. The limited data on humans implied a high variability in the bioavailability of exogenous melatonin ranging from 1% [226] to 100% [227]. Fourtillan et al. have reported an absolute bioavailability with values of 1 and 37% for male and female, respectively. Lane et al. have calculated oral bioavailability from previous data and found it was 0.03–0.06% [228], 0.03–0.76% [229], and 0.09% [230]. Additionally, DeMuro have reported 15% as absolute oral bioavailability of melatonin [225]. This variability is related to remarkable inter-individual variations in all pharmacokinetic aspects including absorption, metabolism, and elimination. These variations highlight the need for further studies on the bioavailability of exogenous melatonin are still needed [231]. In human, exogenous melatonin is well absorbed orally, well distributed, and completely metabolized [232]. It is metabolized extensively by hepatic first pass metabolism with high volume of distribution and cleared by liver [233]. Yeleswaram et al. have investigated the oral bioavailability of synthetic melatonin in animal models showing clear variation in the bioavailability of melatonin. The bioavailability was 53.5% orally and 74% intraperitoneally in rats, and 100% orally in monkeys and dogs [227]. In the same study investigating the in vitro permeability on CACO-2 human cells, melatonin showed good absorption [227]. Aguilera et al. investigated the impact of the aqueous extract intake of bean (*Phaseolus vulgaris* L.) sprouts, a source of phytomelatonin (melatonin of plant origin), in rats. In addition to a comparison between the bioavailability of bean sprouts-derived phytomelatonin and synthetic melatonin, the results showed increased plasma level of melatonin after administration of phyto- and synthetic melatonin with higher bioavailability (17%) of melatonin for synthetic melatonin treated rats [37]. In another study, Andersen et al. have performed a crossover study in healthy volunteers to explore the pharmacokinetics of oral and iv melatonin. Following administration of 10 mg melatonin, orally or intravenously, low absolute bioavailability (3%) was demonstrated with inter-individual variations for oral melatonin, and lower values of Cmax for oral melatonin as compared to iv melatonin [231].

## 8. Melatonin in Clinical Trials

Experimental and clinical studies have determined that melatonin exhibits significant prophylactic properties against the toxic adverse event profiles of chemotherapy and radiotherapy [234,235,236]. It has also been investigated as a complementary modality alongside chemotherapy owing to its antioxidant and immunoregulatory influences. However, in this capacity, melatonin is still the subject of research and not in use in routine clinical practice [237,238]. Several studies have documented that melatonin demonstrates a variety of anti-tumor actions. These encompass antioxidant, cytostatic, anti-proliferative, and pro-apoptotic effects, together with various activities pertaining to its ability to regulate epigenetic responses [236,239,240]. There are increasing data to show that these anti-malignancy traits are evident during several phases of tumor advancement and dissemination [241], although recent reports have indicated that these influences are poor or non-existent [242,243].

The potentiating influence of melatonin on additional anti-tumor agents requires further elucidation in clinical studies. Moreover, its direct impact, utilizing exogenous administration, on individuals with definitive neoplasia requires further study in order to delineate melatonin’s effect on tumor progression and to generate its data profile relating to dosage and adverse events. The compound’s modes of action also require clarification [2].

Several clinical studies have suggested that the efficiency of chemotherapy can be enhanced by the concomitant prescription of melatonin; side effects of the former are also ameliorated. Data also indicate that survival is increased and life quality improved [244,245,246]. It is thought that the capacity of melatonin to scavenge free radicals together with its antioxidant characteristics, are responsible for the improved outcomes [247]. Further investigation into the properties and clinical use of melatonin using in vivo animal experiments and clinical trials are necessary in order to further delineate the prophylactic and clinically relevant actions of the compound when utilized as an adjunct to chemotherapy.

It has been proposed that melatonin’s benefit in mitigating the toxic effects of chemotherapy and its association with aberrant mitochondrial function should be explored using double-blind placebo-controlled trials. It can be expected that a plethora of information will emerge over the next 10 years relating to the way in which melatonin exerts a positive effect in conjunction with anticancer drugs [248]. The development of resistance to therapy, together with the occurrence of tumor spread, means that the investigation of de novo modes of treatment for malignancies is essential.

In one study, researchers reached the conclusion that one-year survival may be enhanced without adverse effects with the use of melatonin; response rates are accelerated when melatonin is used as an adjunct to several routinely employed anti-tumor treatments. Furthermore, it mitigates the toxic effects of chemotherapy and can lessen symptomatology associated with malignancy [249]. A meta-analysis encompassing 21 clinical studies relating to patients with disseminated solid malignancies concluded that the pooled relative risk (RR) for one-year mortality was 0.63 (95% CI = 0.53–0.74; *p* < 0.001). An improved effect was found for stable disease, partial response, and complete response with statistically significant RRs of 1.51, 1.90, and 2.33, respectively. In studies combining melatonin with chemotherapy, adjuvant melatonin decreased one-year mortality (RR = 0.60; 95% CI = 0.54–0.67) and improved outcomes of stable disease, partial response, and complete response; statistically significant pooled RRs were 1.15, 1.70, and 2.53, respectively. In these studies. melatonin also significantly reduced thrombocytopenia, leucopenia, asthenia, nausea, vomiting, and hypotension [249]. Independently conducted well-designed trials are needed to confirm these findings. Table 1 shows the published clinical human studies evaluating the effects of melatonin on cancer patients.

Such research implies that melatonin is not a suitable compound for the initial phase of malignancy treatment; recommendations for use are restricted to combination treatment with mainstay therapeutic drugs and other modalities. Table 3 summarizes latest clinical studies of using melatonin to treat different cancers.

### 8.1. Melatonin as an Adjuvant to Radiotherapy

Emerging literature in the last few years has demonstrated that melatonin used alongside radiotherapy is able to augment the impact of ionizing radiation on tumors and can also prevent the latter’s toxic effects on non-cancerous cells. In vivo and in vitro scenarios have been deployed to examine melatonin’s radio-sensitizing properties; a range of modes of action have been postulated for its activity in this regard [94,194,255,256].

Although multiple in vivo and in vitro studies have been performed in order to evaluate this phenomenon, the use of melatonin to enhance the effect of radiotherapy in human subjects has been poorly studied. Lissoni et al. were one of the first institutions to try to study melatonin’s influence on ionizing radiation; they looked at this combination of treatment, using 60 Gy of radiation, in 30 patients presenting with glioblastomas. Their initial publication indicated that radiotherapy in combination with melatonin in this clinical cohort may improve life quality and increase the one-year survival statistics [257].

This positive outcome was not experienced in a randomized phase II clinical trial evaluating patients with cerebral metastases who were administered 30 Gy radiotherapy in 10 fractions in the afternoon, and were randomized to 20 mg melatonin which was prescribed for morning or evening administration. No clinical value with respect to either survival or neurological tumor progression was discerned [258].

One study reported that an emulsion therapy with melatonin as a constituent notably diminished manifestations of radiation-induced dermatitis in patients in stages I, II, or 0 of breast neoplasia that received 50 Gy radiotherapy to the entire breast [259]. A cohort of individuals, comprising low numbers, who received pelvic radiotherapy for malignancy in the presence or absence of exogenous melatonin, were examined with respect to radiation-induced lymphopenia. Melatonin was found to have no impact on the development of the hematological disorder [260].

A meta-analysis encompassing 21 clinical studies relating to patients with disseminated solid malignancies concluded that melatonin appeared to be helpful in those individuals who were also being treated with chemicals, ionizing radiation, and undergoing supportive or terminal care. Enhanced survival was reported together with mitigation of the toxic consequences of chemotherapy. The review only included one paper relating to the combination of melatonin and ionizing radiation [249].

The ability to potentiate the efficacy of radiotherapy in patients with malignancy could have multiple advantages. The oncostatic actions of melatonin mean that this compound is of great interest in tumor therapy. In particular, the in vitro and in vivo data obtained for melatonin, used as a co-treatment with ionizing radiation, are striking.

### 8.2. Lung Cancer

Lung tumors are a major contributor to malignancy-related deaths in both males and females [198]. Responsible for approximately 13% of all malignancy presentations, lung cancer is the most commonly diagnosed form of neoplasia [198]; it is also the second most often detected in men [261].

One of the most frequently arising respiratory tract tumors, comprising 85% of lung malignancies [262], is non-small-cell lung cancer (NSCLC) [87]. This is aggressive, with an anticipated 5-year survival rate of just under 16% [263]. Modest improvements in survival can be achieved with surgery and combination chemotherapy and radiotherapy [264], although these treatment forms have a significant toxic event profile in relation to non-cancerous tissues, thus restricting their therapeutic potential [181].

Several studies have investigated ways of diminishing the side effects of therapy and augmenting the effectiveness of chemotherapy and radiotherapy by using complementary agents including melatonin, which are well tolerated [181,265]. Changing melatonin’s circadian rhythm has been postulated as causing a rise in the incidence of NSCLC [239]. A few research groups have suggested that melatonin may be a possible treatment option in patients with lung tumors, predominantly owing to melatonin’s ability to potentiate radiotherapy and other anti-tumor agents.

It is possible that melatonin could act as an anticancer agent in the therapy of NSCLC, together with additional forms of neoplasia. Given melatonin’s ability to potentiate radiotherapy and chemotherapy and to minimize their adverse event profiles, its use as an adjunct may facilitate higher doses of the former and thus augment their efficacy to treat malignancy. In addition, since melatonin has anti-proliferative, pro-apoptotic, anti-metastatic, and immunostimulatory properties, it merits more attention as a potential compound that can decelerate tumor progression. Since melatonin per se is well tolerated and combines well with other forms of therapy, more clinical studies which evaluate melatonin, together with elucidating its mechanism at the molecular level in relation to its beneficial effects on tumors, will assist in optimizing the use of this compound in relation to the therapy of patients with NSCLC [239].

One study, comprising 70 patients with late-stage NSCLC, compared combination cisplatin and etoposide with or without the addition of melatonin. When either total or partial tumor response rates were appraised, patients additionally receiving melatonin demonstrated an augmented response to the chemotherapy; additionally one-year survival was similarly prolonged. Moreover, the frequency of myelosuppression, neuropathy and cachexia were notably decreased, suggesting that melatonin improved patient’s abilities to withstand the chemotherapy [266].

Another study appraised the addition of 20 mg daily melatonin in patients with disseminated NSCLC, who were also being treated with cisplatin and etoposide. In those individuals taking melatonin, the overall tumor regression rate and the 5-year survival were elevated, together with an improved clinical tolerance to the pharmaceutical agents [267].

### 8.3. Breast Cancer

Malignancy of the breast is one of the most frequently presenting tumors in females, and one of the main causes of death in this gender in the age-group 40–55 years [268,269].

The majority of breast tumors are generally detected in an early phase of the disease; however, metastases will arise in almost a third of patients even though they receive therapy [270]. Tumor dissemination is pathognomonic of neoplasia and is the main cause of death in breast disease [271]. Pathogenetic mechanisms that explain the incidence of metastatic breast cancer have yet to be fully elucidated at molecular levels; their delineation, however, is essential to future treatments.

There are multiple studies pertaining to the impact that melatonin has on breast malignancy; melatonin has been demonstrated to influence numerous facets of the hormonal system. A spectrum of oncostatic functions related to melatonin has been determined in in vitro studies in breast cancer cell lines, including inhibition of cellular division, invasiveness and stimulation of cellular necrosis [272].

Melatonin has been ascribed with anti-estrogenic activity, and it is, therefore, thought to exert a prophylactic action in relation to breast tumorigenesis. Furthermore, a disturbance in circadian rhythms, e.g., though working nights, might disturb environmental effects on melatonin synthesis. This is a potential risk factor for the development of breast neoplasia [273].

Data from a meta-analysis have implied that melatonin may influence the prevalence of breast tumors in females. Further work is necessary to understand methodological discrepancies [273].

Several researchers have described enhancement of sleep and life quality in individuals with breast neoplasia who were prescribed melatonin. Melatonin, given at night, was reported to give rise to notable benefits in objective and subjective sleep qualities, sleep fragmentation and amount, the degree of tiredness experienced, overall life quality, and increased scores on scales relating to social and mental function [274].

In a double-blind placebo-controlled randomized study, a reduced likelihood of presenting with symptoms suggestive of an affective disorder was reported in individuals administered 6 mg oral melatonin compared with those given a placebo [252]. Reported secondary endpoints in this study included the fact that an evening prescription of melatonin 1 h before sleep enhanced sleep efficacy and diminished wake after sleep onset for the fortnight following surgery [252].

A further study, also randomizing melatonin versus a placebo, noted that participants receiving the former reported benefits in subjective quality of sleep when assessed using the Pittsburgh Sleep Quality Index [250]. The administration of melatonin admixed with somatostatin, retinoids, vitamin D3, and low-dose cyclophosphamide proved to be beneficial with respect to effectiveness and survival statistics in humans with breast malignancy [253] although in a study that, again, had a double-blind placebo-controlled design, melatonin failed to impact estradiol and IGF-1/IGBBP-3 titers in females with a previous history of breast malignancy of stage III or less [251]. Yet another double-blind placebo-controlled crossover study, comprising 72 subjects, obtained results that indicated that 20 mg oral melatonin failed to ameliorate tiredness or related symptomatology in individuals with late-stage carcinoma [254].

### 8.4. Colorectal Cancer

Another global leading cause of death from malignancy is colorectal cancer (CRC) [169,275,276,277]. Melatonin has been demonstrated to have anti-tumor effects for CRC in a number of studies. If detected early within its disease course, CRC has an excellent prognosis with a 5-year survival rate of 90% following definitive surgery; however, this figure drops to 14% once remote metastasis occurs [278].

In CRC, therapeutic strategies include surgery, together with neo-adjuvant and adjuvant chemotherapy approaches, and treatment that is specifically targeted to tumor cells, e.g., antibodies and kinase inhibitors [279]. In patients with disseminated cancerous lesions, such multi-modality options have improved survival to a median value of almost 2.5 years [280]. The mainstay of treatment is surgical tumor excision; when offered together with chemotherapy 5-year survival is 58% [54]. Unfortunately CRC cells are recognized as having the capacity to develop resistance to pharmaceutical anticancer agents through a range of mechanisms [281]. In order to offer superior treatment regimens and increase survival rates in CRC, new encouraging admixtures of chemotherapy and potential adjuvants require additional study.

Experimental and clinical studies have determined that melatonin exhibits significant prophylactic properties against the toxic adverse event profiles of chemotherapy and radiotherapy [234,235]. These properties may enable increasingly powerful and, therefore, more effective chemotherapy regimens to be utilized [57]. Furthermore, melatonin per se also has anti-proliferative, anti-metastatic and cytotoxic properties on a spectrum of human cancers; these results have included work on CRC [2,193,282]. It should be noted that this molecule, which is intrinsically produced, has a benign adverse event profile even when administered in relatively high quantities [34,231,241].

In 1987, Lissoni et al. were the initial workers to report the impact of melatonin in malignancy [283]. They recruited 19 patients with late-stage solid cancers, encompassing some with CRC, who had failed to gain benefit from conventional treatments. In total, 20 mg of intramuscular melatonin was prescribed on a daily basis. In individuals who went into remission, evidenced disease stability, or were able to function more effectively, the melatonin dose was reduced to a maintenance regimen. Improved performance scores and enhanced life quality were reported in 60% of the study subjects. This early work highlighted that melatonin had the potential to be of therapeutic value in patients for whom traditional treatment methods were no longer effective [283].

The clinical impact of melatonin in individuals with disseminated CRC in whom 5-fluorouracil therapy was ineffective was studied by Barni et al. [284]. 20 mg daily melatonin was prescribed and delivered as an intramuscular injection for 8 weeks; maintenance therapy comprised 10 mg oral melatonin. Overall, 5 of the 14 patients in the study demonstrated obvious positive change in their functional abilities, although no anticancer properties attributed to melatonin were described in this study.

One trial enrolled 1440 patients who had late-stage solid tumors, including 279 with CRC, which were deemed to be beyond conventional therapy. In the first section of this study, half of the patients were given melatonin in addition to standard supportive care [285]. The latter phase of the study used melatonin, 20 mg per day given in the evening, alongside chemotherapy, which comprised treatment with 5-fluoroucracil with folinic acid or raltitrexed. This patient group of 200 individuals had disseminated tumors that were resistant to chemotherapy, of whom a quarter had CRC. The data from this work implied that melatonin could act as a prophylactic agent in order to diminish symptomatology related to advancing cancer, e.g., cachexia, asthenia, and lymphocytopenia, together with adverse events arising from the pharmaceutical agents, i.e., thrombocytopenia, asthenia, and cerebral and cardiac toxicity. Mutually potentiating interactions of melatonin with the anti-tumor agents were also noted.

Again, in patients with CRC and distal metastases, one randomized study appraised the concurrent use of melatonin together with irinotecan [286]. A total of 30 patients were included who had failed to show regression despite a minimum of one episode of chemotherapy including 5-fluorouracil. In total, 20 mg oral melatonin was given to the relevant patient cohort in the evening. Those patients experienced a higher degree of disease control, i.e., 85.7% compared with the group that did not receive melatonin, i.e., 43%.

A further clinical trial randomized 370 individuals with malignancy, which included 122 with CRC, to undergo chemotherapy with or without oral melatonin, dosed at 20 mg per day [287]. Patients with CRC had been receiving combination oxaliplatin, 5-fluororuracil and folinic acid, 5-fluorouracil and folinic acid, or irinotecan on a weekly schedule. In those patients in whom melatonin was given as an adjunct, the regression rate of malignancy was elevated, and a prolonged 2-year survival was documented.

In vitro and in vivo studies have been performed, encompassing both experimental animal models and clinical trials in order to appraise the properties of melatonin, either as a sole agent or as an adjunct to anti-tumor strategies. Clinical studies have mostly recruited patients at a late stage of their disease. However, the optimum route and dose of melatonin is yet to be determined; additional studies are required. In order to further elucidate the benefits of melatonin in patients with CRC, animal work in particular is necessary in order to provide a knowledge base to use as a foundation for human studies.

In conclusion, there is substantial data to implicate melatonin in tumorigenesis, development and CRC cell advancement through a variety of modes of action. Additional clinical studies are therefore essential in order to be able to incorporate melatonin as an encouraging novel anti-tumor agent in patients with CRC [278].

### 8.5. Hepatocellular Carcinoma

Worldwide, liver tumors are attributed with being the second most frequent cause of demise from malignancy. Hepatocellular carcinoma (HCC) is the most prevalent cancer type, comprising between 70 and 80% of hepatic lesions. It is most common in less developed nations [288,289]. Surgical resection is the only definitive therapy for HCC. However, many patients fail to meet operative inclusion criteria, and so efficacious chemotherapy regimens need to be established [290].

Over recent years, a remarkable proliferation of the prevalence of liver malignancy has been reported globally; in total, 841,000 instances of liver tumors and 782,000 deaths were reported from this disease in 2018 [169]. This form of malignancy is the 5th most common in men, and the 7th most prevalent in women; it is recognized as fourth on the list in terms of fatality [169]. Thus, it can be anticipated that liver malignancies are currently demonstrating a rapid acceleration in both incidence and mortality [291,292].

There are numerous published studies that have assessed the use of melatonin in patients with liver tumors, including the evaluation of its impact and modes of action with respect to HCC [293]. Mesenchymal stem cells (MSC), arising from bone marrow, express Mel receptors. Thus, melatonin exhibits a spectrum of influences mediated through these receptors on MSCs, encompassing survival prolongation, motility, engraftment, and cellular differentiation. These actions appear to be associated with interplay between the receptors and enzymes within the matrix; MSC homing effects are augmented when an admixture of melatonin and MSCs is pre-administered [294].

In contrast, numerous mechanisms have been the subject of theory with respect to MSC-dependent cancer inhibition, e.g., MSCs, pulsed with micro-vesicles obtained from cancer cells, have demonstrated potentiated anticancer properties in patients with HCC [295]. Earlier studies have shown that melatonin is able to augment the likely beneficial clinical effects of MSCs in a selection of pathologies, including acute kidney injury and metabolic disease, e.g., diabetes. Modes of action for these possible therapeutic roles include stimulation of antioxidative pathways, suppression of the inflammatory response and a decrease in both apoptosis and fibrosis [296,297,298,299].

Of note is that some current studies have described a mutually potentiating effect between melatonin and MSCs for targeting the inflammatory processes associated with HCC [300,301,302] although it should be emphasized that there is still a dearth of clinical data in relation to this phenomenon.

In summary, the admixture of MSCs and melatonin in HCC may offer encouraging clinical outcomes by stimulating resistance to cellular necrosis. This mechanism has been documented in experimental murine models of HCC in which combination therapy led to a reversal of hepatic dysfunction and reduced tumor load when compared to administration of each treatment component as a sole agent. These beneficial effects were also noted when the combination therapy was given together with preconditioning [300,303]. Since only minimal studies have been published in relation to this subject, additional work is crucial in order to investigate the future potential of this treatment combination, to elucidate the underlying modes of action and to delineate the application of targeted stem cell therapy in HCC.

### 8.6. Prostate Cancer

Malignancies of the urological tract, which encompass tumors of the prostate, bladder and kidney, give rise to 12% of cancer-related fatalities globally. Prostate neoplasia is the most common, with an annual incidence of one million, and a yearly mortality rate of 300,000 [304,305]. Worldwide, tumors of the bladder comprise the ninth most prevalent malignancy; the annual incidence and death rates from bladder cancer are about 330,000 and 130,000, respectively [305]. Prostate tumors in men become more prevalent with advancing years.

A link between loss of circadian rhythms or sleep deprivation with prostate lesions has been reported in an epidemiological study systematic review [306]. Additionally, a case-cohort study documented a possible relationship between early morning urine titers of 6-sulfatoxymelatonin (aMT6) and prostate malignancy. Males with values that were less than the median were noted to have four times the risk of late stage or terminal prostate tumors when contrasted with individuals with more elevated aMT6s levels [172]. Melatonin circadian rhythm analysis has demonstrated that serum melatonin titers are decreased in individuals with primary prostate malignancy; this is attributed to diminished activity of the pineal gland rather than being induced by heightened hepatic metabolic breakdown [307].

Conventional treatments have shown no survival benefit in patients with prostate cancer, and therefore further studies are urgently needed in order to design additional efficacious pharmaceutical agents as alternative forms of treatment or to be used in conjunction with standard regimens. Further to enhancing sleep and life quality in patients with malignancy, melatonin delivered as an adjuvant to anti-tumor drugs promotes their effectiveness and improves survival rates [308]. In order to demonstrate the clinical value of melatonin in individuals with prostate cancer, additional research is required.

### 8.7. Ovarian Cancer

Among gynecological carcinomas, ovarian cancer presents with a poor prognosis and is frequently fatal [309]. Due to the lack of early detection methods, this type of cancer is usually presenting in its advanced stage when it is discovered. Therefore, it is important to find new therapeutic strategies. We need first to identify the laboratory and biological sources of inconsistency in melatonin levels determined in samples to investigate the association of melatonin with cancer.

Over a five-year period in the Prostate Lung Colorectal and Ovarian Cancer Screening Trial (PLCO). Serum melatonin levels were measured in 97 participants to test if melatonin levels are constant over time. The results of this study regarding the high correlation of melatonin levels indicates that single measurements may be used to discover population level associations between melatonin and risk of cancer [310].

### 8.8. Brain Tumors

Glioblastoma is the most frequently identified cerebral cancer in adults, and also recognized as the most aggressive. The incidence rate is between 5 and 8 adults in every 100,000; it is responsible for just over half of all detected gliomas. A poor survival rate is associated with glioblastoma; the majority of patients have an average life expectancy of under a year after presentation, mostly owing to recurrence [149]. Currently, the main form of treatment is radiotherapy, together with chemotherapy in the form of temozolomide (TMZ) [311].

An alkylating drug administered orally, TMZ is able to cross the blood-brain barrier and gain access to the cerebrospinal fluid. This ability enhances its anti-tumor activities and prolongs the life expectancy of individuals with glioblastoma [311]. Additional compounds that have been appraised have so far yielded no benefits relating to either overall survival rates or tumor suppression [312].

It is well established that therapy for individuals presenting with glioblastomas is complex; curative surgery is nearly impossible, and the majority of tumors exhibit a high recurrence rate despite treatment with radiation and anticancer agents [313]. Thus, several workers have concentrated on the development of de novo adjuvant treatment approaches, favoring natural compounds in order to offer anti-tumor agents that are suitable for clinical use. A number of studies have documented the properties of melatonin with respect to glioblastomas. Melatonin has well-known antioxidant effects, and its anti-tumor actions are becoming acknowledged. It, therefore, has potential to thwart the resistance to numerous chemotherapy agents that plagues treatment of glioblastomas [159,160,313]. Additional work is necessary to design novel molecular compounds and approaches, combination treatments, and optimal dosing regimens.

Resistance to chemical anti-tumor drugs is a central issue in chemotherapeutic approaches to this tumor. The principal objectives for combination therapies encompassing prolongation of survival rates and enhancing life quality include mitigating the cytotoxic adverse event profiles of pharmaceutical agents and simultaneously diminishing tumor resistance and unwanted drug effects. Studies in experimental and clinical scenarios have demonstrated that melatonin can potentiate the anti-tumor actions of chemotherapy in numerous forms of malignancy. This action, together with diminishing the toxic impact of chemotherapy, enhances life quality [236,312].

Although a few studies have reported anti-tumor actions of melatonin in relation to glioblastoma in vitro, as yet, few animal models have been published, and there is scant literature available on this subject in humans.

An in vivo experiment by Martin et al. documented the efficiency of melatonin administration in diminishing cancer cell growth [314]. Melatonin was used to pre-treat glioma cells, which were then administered to rats parenterally. In contrast to a control cohort, a notable decrease in malignant growth was observed six days after commencement of therapy, i.e., at day 11, reaching 50% by two weeks after treatment initiation. However, the dose of melatonin prescribed, i.e., a subcutaneous injection of 15 mg/kg body weight, was much higher than utilized in human studies, the latter typically being 20 mg orally per day. Thus, in subsequent clinical studies, updated dosing regimens for melatonin, similar to those used effectively in in vivo experiments, may lead to the appreciation of the anti-tumor effects of melatonin as a sole agent in individuals with glioblastomas.

To date, only a single clinical trial has appraised adjunctive treatment with melatonin in patients with glioblastoma. Lissoni et al. [257] studied the additional use of melatonin in 30 patients with this tumor, who were undergoing radical or complementary treatment with ionizing radiation. Randomization to receiving or not receiving oral melatonin, dosed at 20 mg per day, was performed. The addition of melatonin at one year follow-up conferred a survival advantage compared with the control cohort, i.e., 6/14 and 1/16, respectively. This study also documented a reduction in the adverse effects relating to steroid treatment in the melatonin group, including less infections and alopecia, alleviated symptoms of anxiety and enhanced sleep quality. Further studies need to follow up this preliminary data, incorporating larger patient populations. However, this early work highlights that melatonin may be a suitable therapy adjunct in patients with glioblastoma, and have effects on the tumor per se, as well as promoting life quality.

Lissoni et al. published an additional study, investigating treatment with an admixture of melatonin and Aloe vera [315]. The latter is known for its anti-inflammatory properties. The purpose of this trial was to determine whether these two compounds could act in synergy to improve the anti-tumor properties of melatonin. A total of 50 patients with malignancy, including breast, lung, gastrointestinal tract tumors and glioblastomas, and who had developed resistance to chemotherapy, radiation, and hormone treatments, or who were unable to tolerate chemical anticancer agents, were recruited for the study. There was a minimum time period of one month between the final episode of chemotherapy and starting 20 mg per day of oral melatonin and tincture of Aloe vera. The study endpoint was tumor suppression. A total of eight weeks after therapy commencement, no effect on lesion regression was seen in the cohort only receiving melatonin. In the group taking Aloe vera and melatonin, 2/24 patients (8%) exhibited a partial response. The side effect profile for melatonin was benign, but Aloe vera was associated with diarrhea, although this was limited to the initial day of prescription in some patients [315].

One issue is that a melatonin concentration of 1 mM has been used in in vitro studies which may not be pragmatic physiologically. Data from McConnell have indicated that a melatonin concentration of 50 nM is similar to the measured pharmaceutical titers of 54 nM in individuals prescribed 20 mg orally [316]. It, therefore, may not be possible to achieve serum levels of 1 mM.

Data relating to the utilization of melatonin as an adjunct to chemotherapy are encouraging, both in terms of augmenting the effectiveness of therapy and mitigating adverse event profiles [159,316]. However, clinical studies that have investigated the clinical efficacy of melatonin in conjunction with other forms of treatment in patients with neoplasia, excluding glioblastoma, have usually been performed outside evidence-based recommendations following lack of success with conventional therapy and a guarded life expectancy.

Clearly, there are some restrictions with respect to how effective melatonin might be in promoting tumor suppression. Despite in vitro confirmation of anti-neoplastic properties, together with documented palliative benefits in human studies, adding melatonin to first-line therapy for cancer treatment has ethical difficulties, although simply enhancing sleep quality or alleviating the toxic effects of more traditional anticancer treatment strategies are clinical benefits that should not be underestimated [8].

An initial stage with the aim of evaluating the possible utilization of melatonin in patients with glioblastoma, possibly admixed with TMZ, could be the inauguration of a scientific evidence consortium, which appraises both in vivo and preclinical data [312].

### 8.9. Osteosarcoma

Osteosarcoma is identified as high-grade primary bone cancer. Despite the many available chemotherapies, the 5-year survival rate is only around 65%. Melatonin could be presented as a new strategy for the osteosarcoma treatment due to the parallel incidence of its levels and osteosarcoma.

The results of numerous experimental studies stated that melatonin can exert its anticancer activities against osteosarcoma through activation and inhibition various mechanisms for example induction of anti-proliferative, apoptosis, and anti-oxidant effects. Moreover, the ability of melatonin to inhibit the invasion and migration of osteosarcoma cells to different organs is a promising approach to prevent the metastasis of osteosarcoma. In conclusion, melatonin alone or in combination with other agents may be a good choice for osteosarcoma cancer therapy [317], however, clinical trials are lacking.

### 8.10. Gastric and Pancreatic Cancer

The beneficial role of melatonin for the management of many cancers has been widely proposed [2,318,319]. However, the potential benefits of melatonin in the treatment of gastric and pancreatic carcinomas are less well known and it is, therefore, crucial to concentrate research in this area.

## 9. Melatonin Safety Profile

One of the main considerations about the treatment in the field of oncology is the evaluation of the toxic effects and risk of new therapies directed against cancers. Regarding melatonin, it has recently been stated that both physiological and pharmacological concentrations have no significant toxicity even at high doses, and aside from some rare exceptions [320], similar to other natural supplement like resveratrol [321].

Based on human trials and reported use, melatonin seems to have a high safety profile especially when used in appropriate doses and short term. Although the doses used in the published studies are 10–50 mg/d higher than those used for other indications (0.5–5.0 mg/d), none of the studies found any severe adverse effects linked to melatonin; while, melatonin decreased some of the side effects caused from radiotherapy and chemotherapy [249,320].

The most common side effects are excess sedation and somnolence. Because melatonin has immune-boosting effects, caution should be exhibited in the post-organ transplant patients, as this could increase the risk of graft rejection, though more research is needed [322].

According to a systematic review and meta-analysis of 21 clinical studies dealt with solid tumors, melatonin significantly reduced thrombocytopenia, leucopenia, asthenia, nausea, vomiting, and hypotension [249].

## 10. Clinical Pharmacokinetics and Dosing of Melatonin

A systematic review including 22 studies with 359 participants, offered valuable understandings regarding the pharmacokinetics of administered melatonin [223]. This systematic review documented a time to maximal serum and plasma concentration (T_max_) of around 50 min after immediate-release oral formulations of melatonin. The half-life time of intravenous and oral melatonin was approximately 45 min (28–126 min). Bioavailability following oral administration was low (9–33%) with substantial intra-individual variability. It is suggested that the low bioavailability is due to the considerable first-pass metabolism in the liver [323]. Additional systematic review of experimental or clinical studies examined the pharmacokinetics of alternative administration regimen for melatonin [324]. Intranasal administration proved a higher T_max_ and bioavailability compared with oral melatonin (2.5–7.8 min and 55–94%, respectively). While the oral transmucosal regimen achieved higher maximal serum and plasma concentrations with similar T_max_ compared with oral route of administration, transdermal administration generated slow melatonin’s absorption and deposition in the skin. Melatonin showed to be safe for daily doses up to 100 mg/kg [325]. However, the majority of the studies mainly involved healthy participants, but preceding studies implied that the melatonin’s pharmacokinetics is influenced by health status, age, and other factors, such as cigarette smoking, caffeine intake, and oral contraceptives use [231,326].

According to a crossover cohort study that examined the pharmacokinetics of intravenous and oral melatonin in healthy male participants [231], oral melatonin was rapidly absorbed, and T_max_ was reached after 41 min. C_max_ and AUC varied greatly between participants. Elimination half-lives after intravenous and oral melatonin administration were 39 and 54 min, respectively. The bioavailability of oral route of administration was only 3%, but a substantial variability between the participants was observed.

Melatonin is available as immediate-release (1, 3, 5, and 10 mg), and controlled-release oral tablets (3 and 5 mg). A 2 mg sustained-release formulation is available outside of the United States. The usual initial dose is 1 mg once daily, and 0.5 mg in geriatric patients. Melatonin should be administered within one hour of bedtime to mimic the body’s endogenous nocturnal surge. In the case of in liver failure, lower doses should be used due to the extensive hepatic metabolism of melatonin, although specific recommendations have not been published [224,242].

In conclusion, several administration regimens for melatonin have been investigated, but it is not yet clear which regimen results in the optimal pharmacologic effect.

## 11. Pharmaceutical Formulation of Melatonin

Melatonin has short plasma half-life, variable oral absorption, and low variable bioavailability that could be due to extensive first pass metabolism [327], in addition to its poor solubility and stability [328]. Therefore, conventional oral dosage forms (immediate release) are unsuitable candidates for melatonin delivery. To overcome these limitations, many pharmaceutical formulations have been developed using different approaches and different routes of administration. For instance, Martarelli et al. have used the hydrophilic polymers; xanthan gum, hydroxypropyl methylcellulose, and Carbopol^®^ 974P NF to formulate tablets which showed prolonged release of melatonin [329]. Another tablet formula as monolayered and three-layered have been prepared by Vlachou et al. with incorporation of polyvinylpyrrolidone and cellulose acetate as nanofibrous mats loaded with melatonin. These tablets showed prolonged release of melatonin as well [330]. Circadin^®^ tablets, as the only licensed melatonin in United Kingdom, showed prolonged release profile, with no effect on the release profile upon division of the tablets [331]. Proietti et al. have prepared soft gel capsules of melatonin and evaluated the pharmacokinetics in comparison with melatonin powder. A total of 1 mg soft gel capsules of melatonin showed comparable pharmacokinetics parameters (AUC_0-360_, C_max_ and T_max_) to 3 mg melatonin powder and both were significantly have better pharmacokinetic parameters than 1 mg powder, the Cmax values were 2620, 2405, and 799.1 μmol/L, respectively [332]. Recently, Li et al. have used porous starch as a carrier producing melatonin-loaded porous starch to improve the pharmacokinetics of melatonin. Increased suppression to DCFH–DA-oxidized peroxyl radicals have been shown with melatonin-loaded porous starch as compared with raw melatonin. In addition, melatonin-loaded porous starch showed an increase in C_max_ (291.77 and 134.26 ng/mL at 15 and 20 min, respectively), and higher AUC_0-360_ with 2.34 folds in treated groups as compared to raw melatonin [333]. In another study, Li et al. have prepared sustained release enteric melatonin-loaded nanosphere composed from silica and hydroxypropyl methylcellulose to enhance bioavailability of melatonin. As compared to raw melatonin, the prepared melatonin-loaded nanosphere showed an increased Tmax and increased Cmax 168.86 to 383.71 ng/mL. Moreover, higher AUC with 3.5 folds in melatonin-loaded nanosphere compared with raw melatonin [334]. Vlachou et al. designed a drug delivery system composed of the algal sulfated polysaccharide ulvan as a hydrophilic matrix system loaded with melatonin to obtain a modified release in vitro. This ulvan-based tablets showed higher %release profile than that of the marketed melatonin drug Circadin^®^ in gastric pH 1.2 [335]. To deliver melatonin to the brain, intranasal administration is a promising alternative to oral route. In this context, de Oliveira Junior et al. have prepared nanoparticles of melatonin-loaded polycaprolactone as intranasal dosage form. These melatonin-loaded nanoparticles enhanced the solubility of melaonin with 35 fold. Moreover, this formulation resulted in IC50 that is 2500 fold lower than raw melatonin. In addition, the AUCbrain in melatonin-loaded nanoparticles treated glioblastoma cells and targeting index were higher than that of raw melatonin administered orally or intranasally [336]. Another intranasal preparation was developed by Priprem et al. when they encapsulated the melatonin in nanosized niosomes. The intranasal melatonin-loaded nanoniosomes were shown to be bioequivalent to melatonin intravenous injection in rats [337]. Terraneo et al. investigated transdermal route as non-invasive alternative to administrate melatonin followed by cryopass laser treatment on mice. The cryopass laser in the transdermally-melatonin pretreated groups showed the same efficiency as the interperitonial-melatonin pretreated group, showing that transdermal administration of melatonin represent a promising noninvasive route with targeting potential on the site of action [338]. Recently, a US patent have been registered for administration of melatonin sublingually where melatonin is complexed with a valerian extract providing a rapid and complete dissolution of melatonin in saliva. This rapid absorption and rapid onset lead to melatonin pharmacokinetics that is comparable to an intravenous melatonin (patent). Furthermore, Li et al. have investigated the melatonin-loaded bacterial cellulose nanofiber suspension to enhance the solubility and bioavailability of melatonin. This formulation showed higher dissolution rate, and the oral bioavailability was 2.4 times higher that of the marketed melatonin [339]. Another formulation was prepared by Terauchi et al. to improve the solubility of melatonin using complexation approach. An inclusion complex of melatonin with 2-hydroxypropyl β-cyclodextrin was prepared. Upon addition of 2-hydroxypropyl β-cyclodextrin, the solubility of melatonin showed linear increase, in addition to the increased uptake of MC3T3-E1 cells as compared to free melatonin [340].

## 12. Conclusions

The role of melatonin in cancer treatment and prevention have been widely studied and numerous experimental studies proved the anticancer effect of melatonin against many cancers, including colorectal, breast, gastric, prostate. ovarian, lung, and oral. The anticancer effect of melatonin is mediated by integrated mechanisms, such as apoptosis induction, immune system modulation, targeting cancer altered mechanism, angiogenesis inhibition, and antimetastatic effect. Combination of melatonin with conventional anticancer therapies showed positive results through reinforcing the therapeutic effects of these therapies. Clinically, melatonin was active to augment the therapeutic effects of anticancer drugs and improve the sleep and life quality of cancer patients. Several clinical studies have suggested that the efficiency of chemotherapy can be enhanced when melatonin was incorporated. The therapeutic effect of melatonin increases when combined with other anticancer agents. Studies showed that the melatonin anticancer effect is not tissue specific and its therapeutic and preventive properties were reported in cancer arising from different tissues. Melatonin can be obtained from plants as phytomelatonin. Its levels vary according to plant species and plant parts. However, the highest level was found in seeds. Bioavailability and pharmacokinetic properties of exogenous melatonin are still not fully understood. High variability in bioavailability was reported with values ranging from 1 to 100%. Such variable numbers are mainly due to remarkable inter-individual variations in all pharmacokinetic aspects including absorption, metabolism, and elimination. Bioavailability of melatonin deserves further studies to clearly understand the interindividual differences. Overall, the low toxicity, diverse mechanisms of action, and high efficiency of melatonin support its use in cancer prevention and treatment.

## Figures and Tables

**Figure 1 molecules-26-02506-f001:**
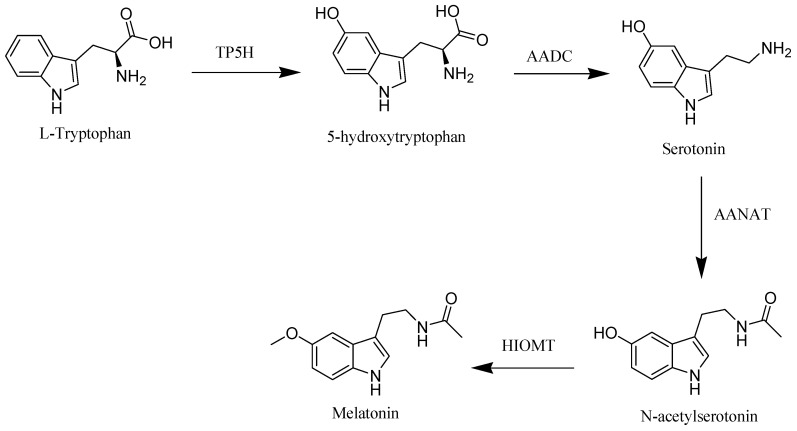
Melatonin biosynthesis in human.

**Figure 2 molecules-26-02506-f002:**
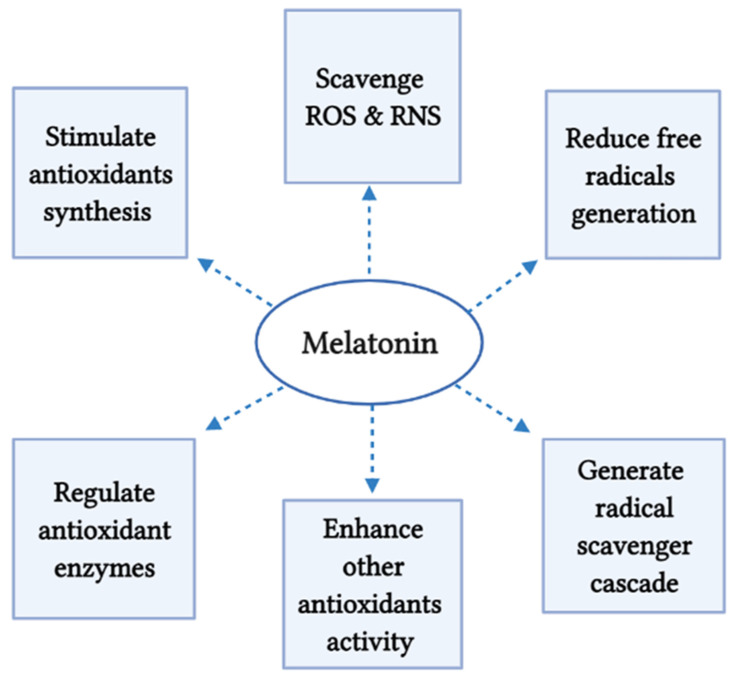
Melatonin’s mechanisms as antioxidant. ROS, reactive oxygen species; RNS, reactive nitrogen species.

**Figure 3 molecules-26-02506-f003:**
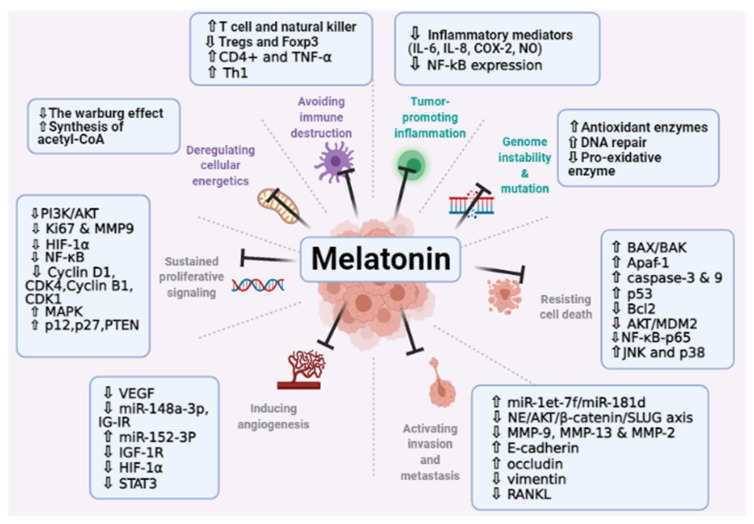
Summary of melatonin activity in restraining cancer hallmarks. BAX/BAK, proapoptotic proteins; NF-κB, nuclear factor kappa B; JNK, c-Jun N-terminal kinase; VEGF, vascular endothelial growth factor; IGF-1R, insulin like growth factor 1 receptor; HIF-1α, hypoxia-inducible factor 1-alpha; STAT3, signal transducer and activator of transcription 3; MAPK, mitogen-activated protein kinase; PTEN, phosphatase and tensin homolog.

**Table 1 molecules-26-02506-t001:** Phytomelatonin content in some plants.

Plant	Organ	Melatonin [ng/g DW] (or FW *)	Reference
Coffee arabica	Bean	6800	[12]
Black pepper	Leaf	1093	[28]
Tomato	Fruit	2.5 *	[12]
Sunflower	Seed	29	[29]
Walnuts	Seed	3.5	[12]
Curcuma	Root	120	[30]
Cherry	Fruit	18 *	[31]
Almond	Seed	39	[12]
St. John’s wort	Flower	4 *	[32]
Strawberry	Fruit	11.3 *	[33]
Cucumber	Seed	11–80 *	[12]
Wheat	Seed	124.7 *	[34]
Pistachio	Seed	233,000	[34]

* Corresponds to FW. DW, dry weight; FW, fresh weight.

**Table 2 molecules-26-02506-t002:** Anticancer activities of melatonin against different cancer types.

Cancer Type	Study Model	Dose of Melatonin	Main Effects of Melatonin and Outcomes	Reference
Gastric cancer	AGS and SGC-7901 cell lines mice	1 mΜ, 2 mΜ, 3 mΜ melatonin 50 mg/kg melatonin	inhibited cell proliferation via the activation of the IRE/JNK/Beclin1 signaling induced the expression of apoptotic and autophagy-related proteins	[214]
	SGC7901 cell line	10^−4^ M melatonin	affected the expression of differentiation relevant factors; the gene expression of endocan was significantly increased and the activity of lactate dehydrogenase and phosphatase was downregulated	[215]
	SGC7901 and BGC823 cell lines	10^−4^ M melatonin	decreased the motility and migration distance, remodeled cells tight junctions, and increased cells adhesion	[216]
	AGS and MGC803 human gastric cell lines	3 mM melatonin	induced apoptosis by upregulating the apoptosis related proteins; Caspase 3, Caspase 9, and downregulating the phosphorylation and expression of upstream regulators MDM2 and AKT	[217]
	SGC7901 gastric cancer cells	2 mM melatonin	inhibited migration, reduced viability, and induced apoptosis upregulated the expression of phosphorylated (p) p38 and c Jun N terminal kinase (p JNK) protein, and downregulated the expression of nucleic p65	[95]
	Mice Murine foregastric carcinoma (MFC) cells	0, 25, 50 and 100 mg/kg melatonin 0, 2, 4, 6, 8 and 10 mM melatonin	inhibited cells proliferation and decreased the tumor volume increased IL-2, IL-10, and IFN-γ expression decreased IL-6 level	[218]
Glioblastoma	Glioblastoma cell lines (U251 and T98G)	0.1–1000 μM melatonin	Reduced cell viability and self-renewal of glioblastoma cells through blocking EZH2-NOTCH1 signaling axis.	[164]
	U87 MG and A172 cell lines	1 mM melatonin	induced autophagy increased the levels of LC3 II, and Beclin 1 upregulation of Bcl-2, the key initiator of autophagy enhanced the apoptosis in glioblastoma cells	[165]
	U251 and U87 glioblastoma cells	1 nM, 1 mM melatonin	blocked the expression of HIF-1α protein and inhibited the expression of vascular endothelial growth factor and matrix metalloproteinase 2 (MMP-2) under hypoxia	[166]
	Human normal neural stem cells hNSC.100	1 μM, 100 μM, 1 mM melatonin	inhibited the proliferation of glioblastoma initiating cells, decreased the clonogenic and self-renewal ability, and downregulated stem cell markers including the transcription factors sox2 oct3/4, nanog, and the transmembrane glycoprotein CD133 decreases the expression levels of de mRNA of these markers	[167]
Prostate cancer	Xenografted LNCaP in mice	1 mg/Kg melatonin	density reduction in the xenograft micro-vessels (lower angiogenesis), and decreased the growth rate downregulated the Ki67 expression, increased the HIF-1*α* expression, and enhanced phosphorylation of Akt	[144]
	Prostate cancer cell line PC-3 cells	1 mM melatonin	upregulated miRNA3195 and miRNA 374b under hypoxia decreased the mRNA expression of angiogenesis related genes including HIF-1α, HIF-2α and VEGF at mRNA level under hypoxia	[68]
	LNCaP and PC-3 prostate cancer cell lines	1 mM melatonin	increased cell toxicity caused by hrTNF-alpha and NF-related apoptosis-inducing ligand (TRAIL) without affecting the action of docetaxel, doxorubicin, or etoposide induced phenotypic changes, and neuroendocrine differentiation	[142]
Lung cancer	CL1-5 and A549cell lines	0.1, 0.3, and 1 mM melatonin	reduced the expression of CD133 in lung cancer cells inhibited PLC, β-catenin, ERK/p38, and Twist signaling pathways to suppress lung cancer stemness	[177]
	CL1-0, CL1-5 and A549 cell lines male SCID mice	0.1, 0.3, or 1 mM melatonin	reduced the lung cancer metastasis reversed the phenotype of epithelial–mesenchymal through twist inhibited Twist/Twist1 expression via MT1 receptor, p38/ERK PLC, and β-catenin signaling cascades	[219]
	SK-LU-1 cell line with PBMC	1 nm, 1 μm and 1 mM melatonin	increased apoptosis and oxidative stress via reduction in GSH, and increased cell cycle arrest	[220]
Ovarian cancer	SKOV3 ovarian cancer cells	3.4 mM melatonin	inhibited proliferation decreased the expression of the proliferation marker Ki67 reduced the ZEB1, ZEB2, vimentin, and snail expression increased E-cadherin decreased the expression of matrix metalloproteinase 9 (MMP9)	[83]
	OVCAR-429 and PA-1 cell lines	0.4, 0.6, and 0.8 mM melatonin	downregulated CDK 2 and 4 which lead to accumulation of OVCAR-429 and PA-1 cells the G1 phase	[187]
	Rats	200 μg/100 g bw/day	decreased the expression levels of proteins involved in important metabolic processes which are associated with energy generation, mitochondrial processes, antigen presenting and processing, hypoxia, endoplasmic reticulum stress, and cancer-associated proteoglycans overexpression of fatty acids binding proteins, ATP synthase subunit β, and heat shock protein	[188]
Colorectal cancer	HCT116 cell line (p53 wild type)	1, 10, 100 μM melatonin	decreased plasma MT1, and increased the nuclear receptor, ROR*α* induced apoptosis and autophagic process decreased cells population in S-phase decreased Trichostatin A-associated cardiotoxicity via inhibition of A- and E-type cyclins, and upregulation of p16 and p-p21 expression promoted G1 phase arrest	[196]
	RKO cell line	1, 2, and 3 mM melatonin	downregulated the levels of Rho-associated protein kinase 2 (ROCK2), p-myosin light chains (p-MLC), and phospho (p)-myosin phosphatase targeting subunit 1 (p-MYPT1) expression increased occluding and ZO-1 expression decreased the levels of p38 phosphorylation supp-ressed the migration of RKO cells	[197]
Oral cancer	SCC9 and SCC25 cells	1 mM melatonin	decreased cell viability in both cell lines inhibited the expression of the genes VEGF and HIF-1α under hypoxia and the expression of the gene ROCK-1 in SCC9 cells	[141]
	SAS and SCC9 oral cancer cell lines Vincristine (VCR)-resistant oral cancer cells; SASV32, SASV16, SCC9V16, and SCC9V32.	0.5–2 mM melatonin.	promoted the autophagy and the apoptosis of VCR-resistant oral cancer cells via p38, AKT, and c-Jun N-terminal kinase (JNK) inhibited ATP-binding cassette B1 and 4 induced apoptosis and decreased the drug resistance in VCR-resistant oral cancer cells via increasing the expression of microRNAs	[204]
Liver cancer	HepG2 hepatocarcinoma cell line	1 mM melatonin	decreased the cell viability and downregulated the expression of proangiogenic proteins VEGF and HIF-1α under hypoxia and in normal state reduced the cell migration and invasion	[207]
	HepG2 hepatocarcinoma cell line	10^−9^, 10^−7^, 10^−5^ and 10^−3^ mol/L melatonin	enhanced apoptosis in HepG2 under ER stress via selective blocking of activating transcription factor 6 (ATF-6) inhibition of cyclooxygenase-2 (COX-2) expression, and decreasing Bcl-2/Bax ratio	[208]
Renal cancer	A498, 786-O, Achn, Caki-1, and Caki-2 cells. Mice	0.5, 1, and 2 mM melatonin 200 mg/kg melatonin	modulated ADAMTS1 independently of the MT1 receptor, affecting invasion and growth ability induced microRNA -181d and microRNA -let-7f targeting the non-3′-UTR and 3’-UTR of ADAMTS1 to inhibit its expression and reduce the invasive in renal cancer cells	[128]

**Table 3 molecules-26-02506-t003:** Published clinical human studies evaluating the effects of melatonin on cancer patients as reported in the last 10 years.

Cancer Type and Staging	Participants	Sample Size	Study Type	Daily Dose	Treatment Intervention Group	Control Group	Duration and Follow-up	Outcome of the Study	Ref.
Breast cancer survivors with a prior history of stage 0-III	Postmenopausal females who had finished active cancer therapy	95	A randomized double-blind placebo-controlled trial	3 mg	Oral melatonin (*n* = 48)	Placebo (*n* = 47)	Sleep, mood, and hot flashes were assessed at baseline and after 4 months	Compared to subjects on placebo, Participants of melatonin group experienced significantly larger improvements in subjective sleep quality with no substantial adverse effects	[250]
Breast cancer survivors with a prior history of stage 0-III	Postmenopausal females who had finished active cancer therapy	95	A randomized double-blind placebo-controlled trial	3 mg	Oral melatonin (*n* = 48)	Placebo (*n* = 47)	4 months	The safety profile of melatonin was perfect without any grade 3/4 toxicity and adherence was high (89.5%). Melatonin did not affect circulating estradiol, IGFBP-3, or IGF-1 levels. The low baseline estradiol levels may have prevented the detection of any additional estradiol lowering effects of melatonin	[251]
Breast cancer	30–75 years females, undergoing surgery and without signs of depression on major depression inventory (mdi)	54	A randomized double-blind placebo-controlled trial	6 mg	oral melatonin (*n* = 28)	Placebo (*n* = 26)	3 months	Melatonin significantly decreases the risk of developing depressive symptoms	[252]
Breast cancer [early stage (60%) and a locally advanced/metastatic stage (40%)]	30–73 years (mean: 51)	20	Retrospective analysis	70 mg	A biological multimodal treatment (melatonin, somatostatin, retinoid, vitamin D3 and prolactin inhibitors) 10 mg in the morning, at midday, in the evening with meals, and 40 mg at bedtime	-	-	An overall clinical benefit was attained in 75% of cases (complete response, 55% and partial response, 20%). The overall survival rate was 71% for metastatic cases.	[253]
*Advanced Non-small cell lung cancer (NSCLC)*	Average age = 56 years	151	A randomized, double-blind, placebo-controlled trial	10 mg *(n = 51)*, 20 mg *(n = 53)*	Oral melatonin	Placebo (*n* = 47)	Assessment of health-related quality of life was completed at baseline, and at 2, 3 and 7 months.	Melatonin in combination with chemotherapy enhances the adjusted health-related quality of life and a slightly significantly improve the score in social well-being. However, it did not affect survival and adverse events of the participants with NSCLC	[238]
Advanced cancer receiving palliative care	Patients aged ≥18 years from the palliative care, had a histologically confirmed stage IV cancer, and who reported feeling significantly tired	72	A randomized double-blind placebo-controlled trial	20 mg	Melatonin for 1 week orally each night, a washout period of 2 days, then crossing over and receiving the opposite treatment for 1 week	placebo	Outcomes were measured using the Multidimensional Fatigue Inventory (MFI-20) and The European Organization for Research and Treatment of Cancer Quality of Life Questionnaire. Physical fatigue from the MFI-20 was the primary outcome.	No significant differences between the melatonin and placebo periods were Observed for physical fatigue, secondary outcomes, or explorative outcomes.	[254]

## Data Availability

Not applicable.

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
