# Peer review of "Melatonin in Cancer Treatment: Current Knowledge and Future Opportunities"

_molecules, 2021, doi:10.3390/molecules26092506_

Round 1

Reviewer 1 Report

The manuscript by Tailb et al. described about Melatonin in cancer treatment. This review manuscript is basis for researchers and physicians to develop new therapies for cancer treatment. Especially, the schematic summary (Fig. 3) of melatonin activity in restraining cancer hallmarks is in good shape and easy to understand even for the researchers/readers who do not directly work with this area of research. I suggest adding more details to the figure legends (Fig.2 and Fig.3) if the journal has no word limit restrictions with figure legends.  The references are up to date. Overall, the review manuscript has been well written and well presented.

Author Response

Thank you for your comments. Please see detailed response in the attached letter

Reviewer 2 Report

The manuscript is interesting. highlights the role of melatonia in cancer.
The manuscript is well organized and analyzes all aspects of melatonin, starting from biosynthesis to the clinic.
I suggest the authors to better describe the other properties of melatonin as well.
In addition, the authors could also add some examples of drugs that have an antitumor action, see for example these manuscripts:

Ahmed S, Khan H, Aschner M, Mirzae H, Küpeli Akkol E, Capasso R. Anticancer Potential of Furanocoumarins: Mechanistic and Therapeutic Aspects. Int J Mol Sci. 2020 Aug 6; 21 (16): 5622

Küpeli Akkol E, Genç Y, Karpuz B, Sobarzo-Sánchez E, Capasso R. Coumarins and
Coumarin-Related Compounds in Pharmacotherapy of Cancer. Cancers (Basel). 2020 Jul 19; 12 (7): 1959.

Tomko AM, Whynot EG, Ellis LD, Dupré DJ. Anti-Cancer Potential of
Cannabinoids, Terpenes, and Flavonoids Present in Cannabis. Cancers (Basel).
2020 Jul 21; 12 (7): 1985.

The authors in the conclusions should give more emphasis to the use of melatonin.

Author Response

Thank you for your comments. Please see attached the detailed response

Reviewer 3 Report

Manuscript ID: molecules-1143532

Title: Melatonin in cancer treatment: current knowledge and future opportunities

Wamidh H. Talib, Ahmad Riyad Alsayed, Alaa Abuawad, Safa Daoud and Asma Ismail Mahmod

Comments and Suggestions for Authors

The manuscript by Wamidh H. Talib et al. is a Review about the involvement of melatonin in different anticancer mechanisms including apoptosis induction, cell proliferation inhibition, reduction of tumor growth and metastases, reduction of the side effects associated with chemotherapy and radiotherapy, decreasing drug resistance in cancer therapy, and augmentation of the therapeutic effects of conventional anticancer therapies. Also describe melatonin biosynthesis, availability from natural sources, metabolism, bioavailability, anticancer mechanisms of melatonin, its use in clinical trials, and pharmaceutical formulation.

General comment

The topic is relevant in its field and the studies discussed in this review will provide foundation for design and develop new therapies to treat and prevent cancer using melatonin.

However, I have some comments and recommendation to the Author in order to improve the manuscript

I recommend reconsidering after major revision before being published:

Specific issues

  1. Maintain the term Phytomelatonin for Plant origin and melatonin to mammalians. line 95 and following
  2. In the Table 1. “Melatonin content in some plants”, 11 of 13 plants included are from de same reference, I suggest to modified it.
  3. The authors mention that melatonin can regulate the activities of several antioxidant enzymes, but they did not describe it, line 160
  4. It should be defined all the abbreviatures like GSH, line 164
  5. The authors mention that many studies confirmed that melatonin at physiological concentration (1nM) exerts antiproliferative actions and induces apoptosis in breast, prostate, and ovarian cancer cells but they did not describe these studies, line 196
  6. The 5.3. (5.2. Melatonin effect on proliferative signaling) section should describe the effect of melatonin effect on proliferative signaling, however, the information is not clear enough, the proteins described should be clearly associated to the induction of the apoptosis process. For instance: “It also attenuated the VEGF production and phospho-ACT/GSK-3β axis signaling” This last idea is more related to the section 5.4
  7. The 5.3. (Melatonin effect on promoting cell apoptosis) section should be revised, mainly the way that molecular apoptosis process is described, for instance, Bax/Bak are also part of the Bcl-2 family or “apoptosis via modulation of p65, p38, and JNK expression” what does it mean? What is the relevance? Also, it should be revised the nomenclature employed: BAD/BAX genes??

Or: “upregulation of p21, p27, and PTEN protein is another way of melatonin to promote cell programmed death in uterine leiomyoma”, since these proteins are related to cell cycle arrest and proliferation rather to apoptosis

  1. The 5.4. (Melatonin effect on angiogenesis process) section, with similar observations. “It was also reported the inactivation of MMP-2 and MMP-9 as well as the suppression of the p38 signaling pathway in breast cancer cells treated with melatonin”; MMP are implicated in metastasis and cell invasion.
  2. The title of the section 6: Melatonin bioavailability and use in cancer prevention have to be modified. There is a mistake in the concept, the Authors describe de use of melatonin as a therapy anti-cancer not for preventing to develop cancer. The same observation for the section 6.1
  3. The sections 6.1.1 to 6.1.9, I suggest improving the presentation of the information. The different ideas should be linked, not just summarized
  4. The same observation for Table 2. “anticancer activities of melatonin against different cancer types”. Moreover, the information on table 2 must be of high relevance, I suggest revising it.
  5. I suggest also revising the text of figure 3. “Summary of melatonin activity in restraining cancer hallmarks”. For instance, BAC/BAK. OR BAX/BAK?
  6. The title of section 8. “Melatonin as an Adjuvant to Radiotherapy” does not reflect the description of the sub-sections 8.1 to 8.9, because all these sub-sections describe the effect of melatonin as adjuvant to chemotherapy, surgery and radiotherapy, in many possible combinations
  7. As described above, the sections 8.1 to 8.9 include the studies of use of melatonin as adjuvant with the standard therapy anticancer: chemotherapy, surgery and radiotherapy in different types of cancer, but it is not clearly described. I suggest improving the organization of all information to make it easily to understand

Author Response

Thank you for your comments and suggestion. Please see attached the detailed response.

Round 2

Reviewer 3 Report

I recognize the effort made for the Authors, I agree with the modifications and

I recommend accepting this manuscript to be published